# 'Mini analysis' misrepresents changes in synaptic properties due to incomplete event detection

Ingo H. Greger[1] and Jake F. Watson[1,2] 

[1] *Neurobiology Division, MRC Laboratory of Molecular Biology, Cambridge, UK*
[2] *Institute of Science and Technology Austria, Klosterneuburg, Austria*

Handling Editors: Katalin Toth & Conny Kopp-Scheinpflug

The peer review history is available in the Supporting Information section of this article (https://doi.org/10.1113/JP288183#support-information-section).

**Abstract figure legend** Summary of the study where simulated recordings (left) were used to characterise the effect of incomplete detection on mini (mPSC) analysis. Recording noise levels (red) determine the detected event amplitude and frequency, while true changes in amplitude can be misrepresented as detected frequency changes (purple). This study presents a method for estimating the event detection limit (blue) and provides recommendations for robust data analysis.

**Abstract** Patch-clamp recording of miniature postsynaptic currents (mPSCs, or 'minis') is used extensively to investigate the functional properties of synapses. With this approach, spontaneous synaptic transmission events are recorded in an attempt to determine quantal synaptic parameters

This article was first published as a preprint. Greger IH, Watson JF. 2024. 'Mini analysis' is an unreliable reporter of synaptic changes. bioRxiv. https://doi.org/10.1101/2024.10.26.620084

or the effect of synaptic manipulations. However, at the majority of brain synapses these events are small, with many undetectable due to recording noise. The effects of incomplete detection were well appreciated in the early years of synaptic physiology analysis, but appear to be increasingly forgotten. Here we sought to characterise the consequences of incomplete detection on the interpretability of mini analysis, using simulated mPSC data to give full control over event parameters. We demonstrate that commonly reported measures such as mean event amplitude and frequency, are misrepresented by the loss of undetected events. Probabilistic loss of small events results in detected event amplitude distributions that appear biologically complete, yet do not reflect the underlying synaptic properties. With both simulated and experimental datasets, we demonstrate that specific changes in event amplitude are primarily detected as changes in frequency, compromising classical biological interpretations. To facilitate more robust data analysis and interpretation, we detail a means for experimental estimation of the event detection limit and provide practical recommendations for data analysis. Together, our study highlights how mini analysis is prone to falsely reporting synaptic changes, raising awareness of these considerations, and provides a framework for more robust data analysis and interpretation.

(Received 20 November 2024; accepted after revision 8 September 2025; first published online 26 September 2025)

**Corresponding author** J. F. Watson: Institute of Science and Technology Austria, Klosterneuburg, Austria. Email: jake.watson@ist.ac.at

### Key points

- 'Mini analysis' (patch-clamp recording of miniature synaptic currents, mPSCs) is widely used to assess synaptic function, relying on detection of spontaneous synaptic events.
- Detection of mPSC events is almost inevitably incomplete, as event amplitudes are close to the level of recording noise – a limitation that was well recognised in earlier literature but is often overlooked today.
- Using *in silico* simulated datasets, this study characterises how incomplete detection distorts reported parameters and the distributions of detected events.
- These effects can routinely compromise biological interpretation of mPSC data, particularly the interpretation of amplitude and frequency changes.
- We present a method for experimental estimation of the detection limit and make practical recommendations for maximally careful interpretation of mini data.

## Introduction

Synaptic transmission is the primary means of intercellular communication in our brain. Across the central nervous system, synapses are highly diverse, with specialised properties at different connections for distinct signalling functions (Jonas & Spruston, 1994; O'Rourke et al., 2012; Salin et al., 1996). In addition, individual connections are highly plastic and able to change strength in response to neuronal activity. This is a potential means for information storage in the brain (Bliss & Collingridge, 1993; Lisman & McIntyre, 2001; Martin et al., 2000; Nicoll, 2017). For these reasons, research has been performed for decades to determine the functional properties and molecular mechanisms of synaptic transmission.

Synaptic transmission is quantal, occurring through release of discrete packets (vesicles) of neurotransmitter across the synaptic cleft. These properties were first identified at the neuromuscular junction, where large 'endplate potentials' were shown to be made up of smaller 'quanta' of defined size (del Castillo & Katz, 1954; Fatt & Katz, 1952). These ideas were extended to central synapses of the spinal cord (Kuno & Weakly, 1972), where the properties of synaptic transmission are more complex (Edwards et al., 1976a; Jack et al., 1981; Redman, 1990; Redman & Walmsley, 1983). Recordings from individually stimulated axons demonstrated key properties that we know today: that quantal size and release probability vary between individual synapses (Edwards et al., 1976b; Jack et al., 1981; Walmsley et al., 1987), and even between individual release sites of the same cell pair (Edwards et al., 1976a; Redman & Walmsley, 1983). These findings were applied to hippocampal synapses, where discrete synaptic quanta can be observed, but with substantial variability in

the properties of transmission between connections (Hess et al., 1987; Sayer et al., 1989, 1990).

Quantal analysis through stimulation of individual axons remains the clearest means to determine the properties of synaptic transmission, yet is technically challenging to achieve. For this reason, recording spontaneous synaptic events has become a much more widely used approach to determine functional synaptic properties. Using whole-cell patch-clamp of individual neurons spontaneous synaptic input occurring anywhere across the dendritic tree can be recorded to provide an ensemble measure of synaptic properties on a given cell (Bekkers et al., 1990; Brown et al., 1979; Isaacson & Walmsley, 1995; Malgaroli & Tsien, 1992; Zhang & Trussell, 1994). Spontaneous events (spontaneous postsynaptic currents (sPSCs)) are the result of action potential-dependent neurotransmitter release from any cell connected to the recorded neuron, whereas 'minis' (miniature PSCs/mPSCs: excitatory – mEPSCs, inhibitory – mIPSCs) are recorded during action potential blockade (typically through tetrodotoxin (TTX) application) and occur by spontaneous release of individual pre-synaptic vesicles (Kavalali, 2015). Given the ease of patch-clamp recording from individual neurons m/sPSC analysis is routinely employed by labs worldwide to determine synaptic properties or synapse-level effects of experimental manipulations (e.g. gene knockout (Matt et al., 2018; Varoqueaux et al., 2002), synaptic protein manipulation (Gutierrez-Castellanos et al., 2017; Watson et al., 2017) or neuromodulatory action (Choy et al., 2018; Smith et al., 2005)).

Due to the quantal basis of synaptic transmission, mini analysis is often reduced to measurements of event amplitude and frequency, interpreted respectively as post-synaptic strength and presynaptic release dynamics. This oversimplification will almost certainly drive incorrect biological conclusions. Multiple factors give rise to the distribution of recorded minis. First, the high variability in synaptic properties between different connections will be pooled in this ensemble measure (Bekkers et al., 1990; Edwards et al., 1976b). Second, events will be strongly affected by their dendritic location and neuronal properties, with cable filtering reducing the size and slowing the kinetics of more distal synapses (Bekkers & Stevens, 1989; Brown et al., 1981; Jack & Redman, 1971; Rall et al., 1967). Finally, biological changes at both pre- and postsynapse can influence both the amplitude and frequency of mPSCs. For example post-synaptic unsilencing can increase event frequency with no presynaptic change (Isaac et al., 1995; Liao et al., 1995); changes in vesicle content can determine quantal size through presynaptic means (Shi et al., 2022); and changes in vesicle-receptor alignment could change both frequency and amplitude of synaptic events through co-ordinated pre-post mechanisms (Biederer et al., 2017; Scheefhals & MacGillavry, 2018).

Not only can the biological interpretation of mPSC data be easily mistaken, but empirical interpretation of event distributions can be similarly misunderstood. At the majority of brain synapses mini amplitudes are small (0–50 pA) (Bekkers et al., 1990; Sayer et al., 1990) and follow a positively skewed lognormal distribution (Bekkers et al., 1990; Brown et al., 1979; Derkach et al., 1983; Sahara & Takahashi, 2001; Zhang & Trussell, 1994). Patch-clamp noise levels are comparatively large (typically 2–10 pA max to min values/1–5 pA standard deviation when low pass filtered to 10 kHz), and as a result many synaptic events lie beneath the noise (Brown et al., 1979; Isaac et al., 1996; Malgaroli & Tsien, 1992; Mennerick & Zorumski, 1995; Wang et al., 2024). The resulting incomplete detection has serious consequences for interpretation of synaptic properties. Despite being acknowledged since the first recordings of miniature events in hippocampal slices (Brown et al., 1979; Redman, 1990) and having been carefully considered in early years of mini analysis (Diamond & Jahr, 1995; Manabe et al., 1992; Mennerick & Zorumski, 1995; Yamada & Tang, 1993), with the widespread adoption of mini analysis for synaptic physiology, awareness of these effects and their importance has been increasingly forgotten.

Here using *in silico* mPSC simulations we explore the influence of incomplete event detection on interpretation of mini analysis data. Calculating and considering the detection limit for mPSC events is essential for correct interpretation of these recordings for multiple reasons. First, we show that events below the detection limit are probabilistically detected dependent on their amplitude, giving rise to a 'false' distribution that misrepresents modal values. Second, we show that average event amplitude and frequency are not discrete parameters. Changes in event amplitude are predominantly represented as a selective change in mPSC frequency. Finally, using experimentally recorded data we demonstrate a method for estimating the detection limit, allowing more reliable interpretation of recorded data. Together this study characterises the major risk of mPSC misinterpretation, facilitating more accurate future investigation and re-examination of existing datasets.

## Methods

### Ethical approval

Experiments conducted in the UK are licensed under the UK Animals (Scientific Procedures) Act of 1986 following local ethical approval. All procedures were carried out according to institutional and national guidelines, performed under project licence (PPL)

70/8135 in accordance with UK Home Office regulations and approved by the institutional ethical review board.

## Animals

C57BL/6JOlaHsd mice (Harlan/Envigo; RRID:IMSR_ENV:HSD-057) were housed with food and water *ad libitum* on a 12-h light/dark cycle at room temperature (20°C–22°C) and 45%–65% humidity. Animals were killed at postnatal day 6–8 by rapid decapitation without anaesthesia, according to local and national ethical approval.

## mPSC event simulation

Event simulation and recording noise generation were performed following Pernía-Andrade et al. (2012). All simulated recordings were generated in MATLAB. mPSCs were simulated as biexponential functions consisting of a rising exponential ($\tau_{rise}$) and a decaying exponential ($\tau_{decay}$) following the equation below:

$$\text{mPSC} = \left(1 - e^{\frac{-t}{\tau_{rise}}}\right) \times e^{\frac{-t}{\tau_{decay}}} \text{ for } t > 0$$

where $t$ spanned 70 ms (Fig. 1A). $\tau_{rise}$ and $\tau_{decay}$ were randomly selected for each event from a lognormal distribution of realistic possible values ($\tau_{rise}$ $\mu$: 0.2, $\sigma$: 0.25; $\tau_{decay}$ $\mu$: 1.7, $\sigma$: 0.4), with $\tau_{rise}$ constrained between 0.3 and 2.5 ms and $\tau_{decay}$ between 1 and 25 ms. Resulting curves were scaled to peak amplitudes randomly sampled from a lognormal distribution of realistic peak amplitudes (peak $\mu$: 1.6, $\sigma$: 0.8). For scaled amplitude datasets, scaling factors (amplitude addition or multiplication) were applied to target peak amplitudes prior to curve scaling. To study the effect of kinetics on event detection, $\tau_{rise}$ and $\tau_{decay}$ were fixed for all events within each simulation condition, with a randomly

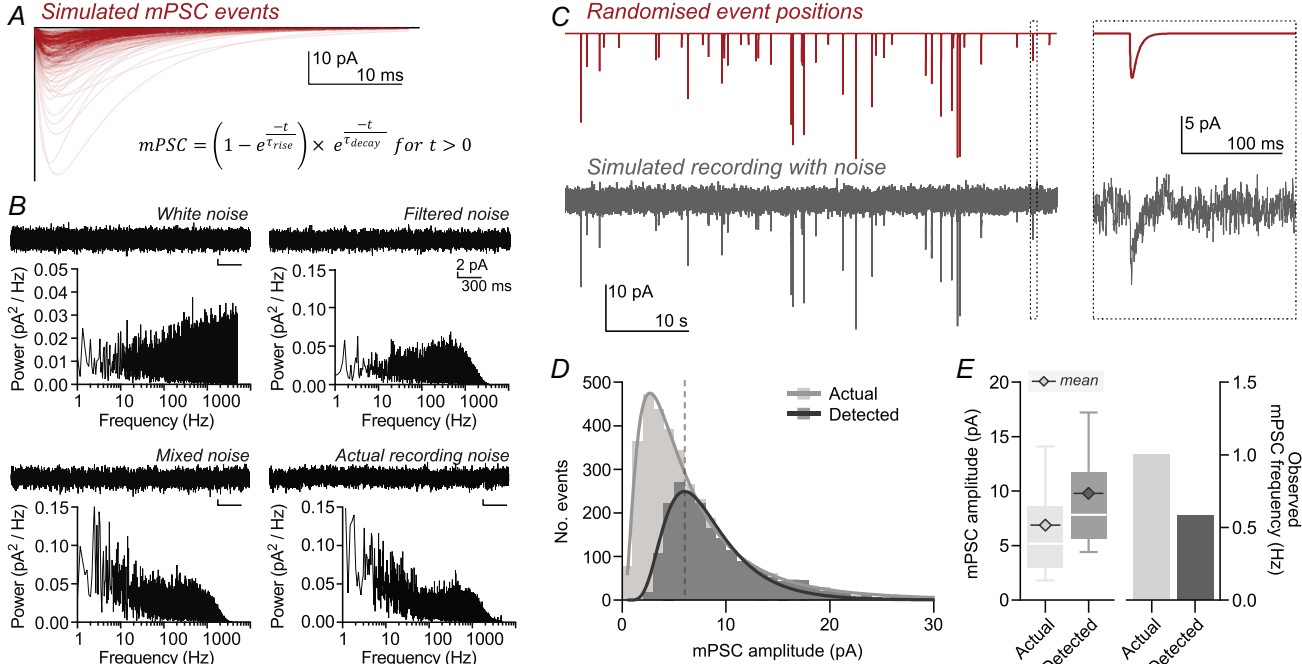

**Figure 1. Incomplete event detection misrepresents measured mini parameters due to loss of small events**

*A*, simulated miniature postsynaptic current (mPSC) events (red) were created from a biexponential function (detailed), with randomised, realistic $\tau_{rise}$, $\tau_{decay}$ and peak amplitudes. A sample of 300 events is depicted. *B*, simulated recording noise was generated from white noise (upper left) by filtering at 1000 Hz (upper right) and incorporation of a 1/f component (lower left). Both simulated noise traces and frequency power spectra are presented, alongside experimentally recorded patch-clamp recording noise (lower right). *C*, events were randomly distributed across a 'time' axis with controlled frequency (upper, 1 Hz) and embedded in simulated recording noise (lower). An expanded view (right) depicts a single event from the wider recording (boxed). *D*, frequency distributions of simulated events before noise embedding (actual) and those extracted by template fit event detection (detected) demonstrate loss of small events after detection. Histogram bars have 1 pA bin width, and the continuous line depicts a lognormal fit of binned data. The dashed line indicates the estimated detection limit calculated using the event scaling method (6.04 pA; see Methods). *E*, the detected mean mPSC amplitude was overestimated (actual: 6.88 pA; detected: 9.79 pA), whereas event frequency was underestimated (actual: 1 Hz; detected: 0.58 Hz; simulated input was 1 Hz events for 1 h duration).

selected distribution of amplitudes as above. In this case, $t$ was extended to 140 ms to minimise clipping of events with very slow kinetics.

Biologically relevant synaptic scaling was implemented by multiplying individual peak amplitudes by a scaling factor dependent on the initial amplitude. This scaling factor followed an exponential decay relationship as follows:

$$\text{Potentiated amplitude} = A \left(s.0.9^A + 1\right)$$

where $A$ is the initial amplitude, and $s$ is the scaling factor of potentiation (varied between 0 and 3).

After their generation, events were assigned locations on the recording trace with uniform randomness, with the total number of assigned events determined from the product of desired frequency and recording length. This 'noise-free' trace was then embedded in simulated recording noise. Events were simulated at 1 Hz unless otherwise stated. Generation of random locations or values was performed using MATLAB functions, and scaled events were given the same position as original events in a parallel recording, therefore creating equivalent simulated recordings aside from scaling of input events.

### Recording noise simulation

Simulated patch-clamp noise was produced by first generating 'white noise' of variable standard deviation (SD) from normally distributed numbers centred on zero, before inclusion of a 1/f 'pink noise' component and Gaussian filtering of the resulting mixed noise at 1000 Hz (Fig. 1*B*). This analysis used existing '1D Gaussian lowpass filter' code (William Rose, 2006; MATLAB File Exchange, File ID: 12 606), and pink noise was simulated following Smith (2011). The resulting 'mixed' noise has a comparable frequency spectrum to 'real-world' patch-clamp recording noise (Fig. 1*B*). The standard deviation of final simulated noise was 2.08 pA for event scaling analysis and 1.47 pA for kinetic analysis.

### Event detection

Event detection employed a standard template search approach, based either on Clampfit or on MATLAB. MATLAB template detection employed the 'minidet' function from the Biosig toolbox (Vidaurre et al., 2011) based on Jonas et al. (1993). Detected events were extracted from recordings or simulations and fit with a biexponential function consisting of a rising and decaying phase, and peak amplitude was taken as the minimum of the curve fit. This approach prevents error in peak measurement caused by recording noise, which is large

for small events close to the noise level (e.g. mPSCs). The equation of fitted curve was:

$$y = a \cdot \left(-e^{-\frac{t-d}{\tau_{rise}}} + e^{-\frac{t-d}{\tau_{decay}}}\right) \quad for\ t > d$$
$$y = 0 \quad for\ t < d$$

where $a$ is the peak scaling factor, and $d$ is the event onset time. mPSC frequency was calculated as the number of events detected per unit time (Hz). The fraction and properties of false-negative ('missed') events were calculated by comparing detected event times with encoded event times.

The detection limit was estimated by sampling datasets of events recorded at either $-70$ mV (scaled by 1.29) and $-90$ mV holding potentials with a sliding bin of 5 pA width at a resolution of 0.1 pA (referred to as 'event scaling method' in figure legends). Using the resulting frequency curves, the $-70$ mV$_{scaled}$ dataset was subtracted from the $-90$ mV dataset, and the frequency difference was plotted against the sliding bin lower limit. This graph was fit with a 'broken stick' curve, where:

$$y = -mx + c \quad for\ x < d$$
$$y = 0 \quad\quad\quad for\ x \geq d$$

and 'd' estimates the minimal amplitude at which no false negatives are recorded or the 'detection limit'.

### Organotypic culture

Organotypic slice cultures were made using the Stoppini method (Stoppini et al., 1991), as described in Watson et al. (2017). Hippocampi from P6-8 mice of either sex were isolated in high-sucrose Gey's balanced salt solution containing (in mM): 175 sucrose, 50 NaCl, 2.5 KCl, 0.85 $NaH_2PO_4$, 0.66 $KH_2PO_4$, 2.7 $NaHCO_3$, 0.28 $MgSO_4$, 2 $MgCl_2$, 0.5 $CaCl_2$ and 25 glucose at pH 7.3. Hippocampi were cut into 300 μm thick slices using a McIlwain tissue chopper and cultured on Milli-cell cell culture inserts (Millipore Ltd) in equilibrated slice culture medium (37°C/5% $CO_2$). Culture medium contained 78.5% Minimum Essential Medium (MEM), 15% heat-inactivated horse serum, 2% B27 supplement, 2.5% 1 M HEPES, 1.5% 0.2 M GlutaMax supplement, 0.5% 0.05 M ascorbic acid, with additional 1 mM $CaCl_2$ and 1 mM $MgSO_4$ (all from Thermo Fisher Scientific; Waltham, MA). Medium was refreshed every 3–4 days, and recordings were performed at 10–12 days *in vitro*.

### Electrophysiology

Hippocampal slices were submerged in room temperature aCSF containing (in mM): 125 NaCl, 2.5 KCl, 1.25 $NaH_2PO_4$, 25 $NaHCO_3$, 10 glucose, 1 sodium pyruvate, 4 $CaCl_2$ and 4 $MgCl_2$ at pH

7.3 and saturated with 95% $O_2$/5% $CO_2$. Excitatory events (mEPSCs) were isolated by addition of 1 μM tetrodotoxin, 10 μM SR-95 531 (Gabazine) and 100 μM D-(-)-2-amino-5-phosphonopentanoic acid (D-APV) (all sourced from Tocris Bioscience).

Borosilicate pipettes (3–6 MΩ tip resistance when filled with intracellular solution) were filled with an intracellular solution containing (in mM): 135 $CH_3SO_3H$, 135 CsOH, 4 NaCl, 2 $MgCl_2$, 10 HEPES, 4 $Na_2$-ATP, 0.4 Na-GTP, 0.15 spermine, 0.6 EGTA, 0.1 $CaCl_2$ at pH 7.25. CA1 pyramidal neurons were recorded in the whole-cell patch-clamp configuration. Signals were acquired using a Multiclamp 700B amplifier (Axon Instruments) and digitized at 10 kHz using a Digidata 1440 A interface (Axon Instruments). Recordings were performed with voltage clamp command potentials of −60 and −80 mV. The liquid junction potential of the bath solution relative to pipette solution was calculated to be +10.1 mV using LJPcalc (RRID:SCR_025044) (Marino et al., 2014). The holding potential values cited throughout the results section are corrected for the liquid junction potential using this value. Series resistance was constantly monitored using a −10 mV pulse every 100 s. Recordings during which the series resistance varied by more than 20% or exceeded 20 MΩ were discarded. The mean series resistance across the mPSC recording time period was 12.9 ± 2.5 MΩ and 14.1 ± 2.9 MΩ for −70 mV and −90 mV datasets, respectively (range: 9.8–16.1 MΩ (−70 mV), and 10.7–18.7 MΩ (−90 mV)).

### Statistics, data analysis and visualisation

All data were analysed in MATLAB (R2022), plotted in GraphPad Prism 9 and presented using Affinity Designer 2. Box and whisker plots present median (line), 25–75 percentiles (box) and 10–90 percentiles (whiskers), overlaid with a symbol at the mean value. Bar plots depict mean values. Values were reported as mean ± SD as specified. Experimental data were presented as individual data points with paired relationships between recordings from the same cell. Statistical comparisons for experimentally recorded datasets were made using Wilcoxon matched-pairs signed-rank test with exact *P*-values displayed on figures. Statistical comparisons were performed using GraphPad Prism 9.

### Results

To determine the effects of the detection limit on recorded mPSC distributions we simulated mPSC recordings with realistic patch-clamp noise (see Methods). Using simulated data allows full control of both mPSC properties and recording noise levels. We first sought to determine the effect of event amplitude on detection around the detection limit. mPSC events were simulated with varying rise time constants, decay time constants and peak amplitudes following a lognormal distribution, approximating real-world data (Fig. 1*A–C*) (Bekkers & Stevens, 1989; Brown et al., 1979; Pernía-Andrade et al., 2012; Sahara & Takahashi, 2001; Zhang & Trussell, 1994). These events were randomly positioned across a simulated recording (1 Hz event rate) and embedded in mixed noise (Fig. 1*C*). We next detected events in these simulated recordings using a standard, template-fit algorithm (Clements & Bekkers, 1997; Jonas et al., 1993). The amplitude of detected events also followed a lognormal-like distribution; however a large number of events were not detected (Fig. 1*D*). Unsurprisingly, undetected events were of small amplitude, hidden in recording noise. As logically expected due to loss of small events, the mean mPSC amplitude was overestimated by detection through recording noise (Fig. 1*E*, actual mean event amplitude: 6.88 pA; detected mean amplitude: 9.79 pA), whereas mPSC frequency was underestimated (Fig. 1*E*, actual frequency: 1 Hz; detected frequency: 0.58 Hz). With complete detection the mode (peak) of mPSC distributions has been suggested to represent synaptic quantal size (Gordleeva et al., 2023; Sahara & Takahashi, 2001). Variability in synaptic properties and dendritic filtering will likely preclude this possibility (described thoroughly in Redman (1990)). In addition to this, the detected modal value in our example was more than twofold overestimated (actual mode: 2.65 pA, detected mode: 5.92 pA). Therefore, incomplete detection produces mPSC datasets that misrepresent underlying physiological parameters.

We next modulated the detection limit by increasing the standard deviation of simulated noise. This caused a shift in the distribution of detected mPSCs (Fig. 2*A* and *B*) and increased the misrepresentation of both mean mPSC amplitude and frequency (Fig. 2*C*). Strikingly, just a 0.6 pA increase in the standard deviation of recording noise caused a 21% increase in the measured mean amplitude (9.9 to 12.0 pA mean amplitude) and a 30% decrease in recorded frequency (0.57 to 0.40 Hz frequency) (Fig. 2*C*). By calculating which simulated events were missed, we determined the association between mPSC amplitude and its likelihood of detection (Fig. 2*B*). Event detection falls away probabilistically with decreasing amplitude, following a sigmoidal relationship. This probabilistic detection of small events shapes the rising phase of the detected event histogram (Fig. 2*A*). Despite missing over a third of events, detected event histograms had a lognormal-like distribution, reminiscent of a complete distribution (Fig. 2*A*). To demonstrate that this curve shape is independent of the input distribution we detected events from a uniform distribution of simulated mPSCs

(Fig. 2*D*), which also demonstrated a gradual loss of mPSCs as their amplitude fell to 0 pA. Due to the overabundance of small events, errors in measured event amplitude and frequency were much larger for a lognormal than a uniform input distribution (Fig. 2*E*). Therefore the nature of synaptic properties unfortunately lends itself to more noise-affected analysis. Crucially, the level of recording noise has a dramatic effect on the observed synaptic parameters. In real-world recordings, noise is not just a result of recording set-up, but it can vary depending on the cell type, cell state, or even within and between individual recordings due to cell health, seal integrity, and recording quality. Therefore it is essential to assess the recording noise of all cells across datasets when comparing experimental groups with mini analysis.

## Detecting changes in synaptic properties

We next sought to determine how *changes* in mPSC properties are observed through the lens of recording and detection. Theoretically, a pure effect on event amplitude would produce a shift or scaling of the amplitude distribution along the *x*-axis, whereas frequency changes would induce a scaling of the distribution in *y* (Fig. 3*A*). Therefore in theory, visualising recorded event distributions should allow simple understanding of underlying synaptic changes. Indeed this is certainly the case when all events are detected, for example large events which occur well beyond the detection limit (e.g. action potential-dependent EPSCs at the cochlear nuclei or Calyx of Held (Sahara & Takahashi, 2001; Zhang & Trussell, 1994). However at the majority of synapses, and

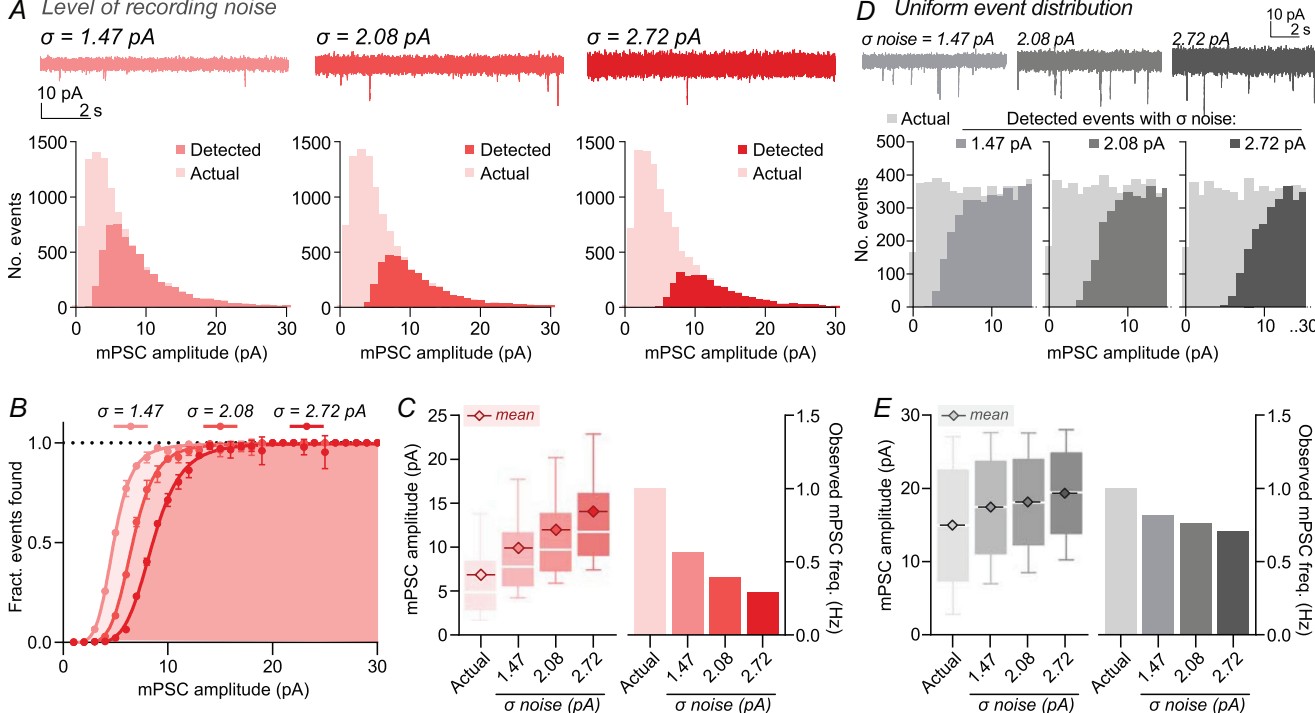

**Figure 2. Probabilistic detection of small events gives rise to a 'false' lognormal event distribution**
*A*, increasing SD of noise (σ) decreased the proportion of detected events (upper, example traces; lower, histograms of actual and detected events, with 1 pA bin widths). Detected event distributions showed a lognormal-like distribution regardless of the fraction of actual events detected. *B*, event detection follows a sigmoidal relationship, with a probabilistic decrease in event detection for smaller event amplitudes. The fraction of events found was fit with a sigmoidal relationship. Symbols and errors depict mean and SD of three simulation repeats. *C*, higher recording noise increased the mean detected event amplitude (actual, 6.8 pA; 'σ = 1.47', 9.9 pA; 'σ = 2.08', 12.0 pA; 'σ = 2.72', 14.1 pA) while decreasing frequency (actual, 1 Hz; 'σ = 1.47', 0.57 Hz; 'σ = 2.08', 0.40 Hz, 'σ = 2.72', 0.29 Hz). *D*, detection of events from a uniform distribution of simulated input amplitudes demonstrates probabilistic detection of small events rather than a strict cut-off. Increasing recording noise shifts this distribution of detected events to higher amplitudes. Tested input amplitudes ranged from 0 to 30 pA, but graphs depict values between 0 and 15 pA for maximal visibility. Plotted histograms use a 1 pA bin width. *E*, with higher noise mean event amplitudes increase and frequencies decrease, but to a lesser extent for uniform than lognormal event distributions, due to the high abundance of small events for lognormal input distributions (mean amplitudes; actual, 15.0 pA; 'σ = 1.47', 17.4 pA; 'σ = 2.08', 18.2 pA; 'σ = 2.72', 19.3 pA. Frequency; actual, 1.00 Hz; 'σ = 1.47', 0.82 Hz; 'σ = 2.08', 0.76 Hz; 'σ = 2.72', 0.71 Hz).

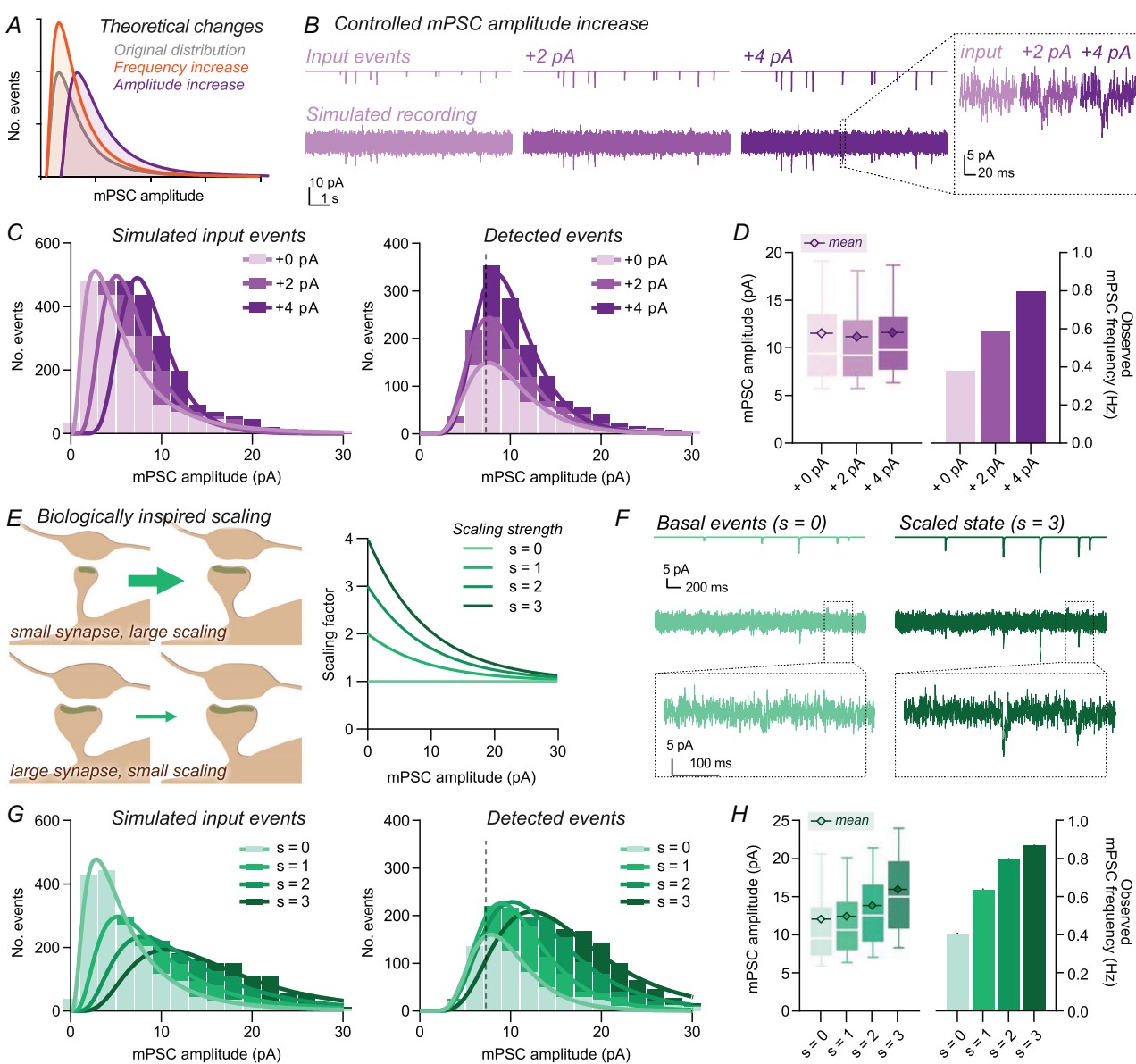

**Figure 3. Specific increases in event amplitude are primarily detected as changes in frequency due to emergence of events from below the detection limit**

*A*, theoretical changes in miniature postsynaptic current (mPSC) amplitude distributions for pure changes in frequency (orange) or amplitude (purple). *B*, example traces of simulated mPSC recordings, with increasing event amplitudes (+2 pA or +4 pA). *C*, simulated input events demonstrated an expected *x*-axis shift by increasing all event amplitudes, whereas distributions of detected events did not display *x*-axis shifting despite pure amplitude modification. Histograms use a 2 pA bin width; continuous lines present lognormal fit of binned data, and the dashed line indicates calculated detection limit using the event scaling method (7.28 pA; see Methods). *D*, mean detected event amplitudes were not altered when increasing the amplitude of input events (mean amplitudes; '+0 pA', 11.5 pA; '+2 pA', 11.2 pA; '+4 pA', 11.6 pA), whereas large changes in frequency were observed (right; '+0 pA', 0.38 Hz; '+2 pA', 0.59 Hz; '+4 pA', 0.80 Hz). *E*, schematic (left) and model (right) of biologically inspired scaling rule, where small synapses were more strongly increased than larger synapses. Tested scaling factors (s) are depicted, where s = 0 corresponds to the unmodified distribution. *F*, representative event simulations, depicting embedded events (upper), and simulated recording (lower) demonstrating basal (*s* = 0) and scaled recordings (*s* = 3). *G*, input event distributions are shown (left). Detected events from simulated recordings (right) showed an increase in the distribution peak at low scaling (*s* = 1), before *x*-axis shifts were also observed with strong scaling (*s* = 3). Histograms use a 2 pA bin width; continuous lines present lognormal fit of binned data, and the dashed line indicates calculated detection limit (7.28 pA). *H*, mean detected mPSC amplitude only increased substantially for strong scaling factors, whereas observed mPSC frequency increased across all datasets (mean amplitude: '*s* = 0', 12.1 pA; '*s* = 1', 12.4 pA; '*s* = 2', 13.9 pA; '*s* = 3', 16.0 pA; observed frequency: '*s* = 0', 0.4 Hz; '*s* = 1', 0.64 Hz; '*s* = 2', 0.80 Hz; '*s* = 3', 0.87 Hz).

for almost all single-vesicle-induced mPSCs, this is not the case.

We repeated our mPSC simulations, but modified the event template to increase the peak amplitude of every event either by 2 or 4 pA before noise embedding (Fig. 3*B*). This manipulation simulates a specific scaling in mPSC amplitude, independent of original synaptic weight. As expected, distribution histograms for simulated events shift along the *x*-axis (Fig. 3*C*). Histograms of *detected* events, however, do not. We instead observe a stretching of the event distribution along the *y*-axis, which is reminiscent of a pure increase in mPSC frequency (Fig. 3*C*). Despite simulating a pure increase in mPSC amplitude, the frequency of detected events increased, whereas mean amplitudes were comparable (Fig. 3*D*). Due to incomplete detection, small events emerge from the noise both to increase the detected frequency and counteract the amplitude increase of previously detected events. Our data demonstrate that mPSC frequency and amplitude are not independent variables when working close to the detection limit. As a result, specific effects on either mPSC frequency or amplitude cannot be reliably reported, and definitive conclusions can only be made from datasets with complete event detection or through analysis of recorded distributions.

We sought to corroborate these findings using a more biologically realistic scaling model. Events were scaled by a factor inversely proportional to their initial amplitude, simulating saturating potentiation and maximal effects on small synapses, as observed in biological systems (Kaneko et al., 2011) (Fig. 3*E* and *F*). With weaker scaling, specific changes to detected event frequency again were observed (Fig. 3*G* and *H*). Only with more robust scaling was a shift in the detected event distribution towards higher amplitudes evident (Fig. 3*G*), yet this was accompanied by a >2-fold increase in mPSC frequency (Fig. 3*H*). These results further confirm the interdependence of mPSC frequency and amplitude for noise-embedded events, as well as the strong sensitivity of mPSC frequency measurements to underlying changes in amplitude.

## Detecting changes in event frequency

To complete this analysis we applied controlled changes in mPSC frequency (Fig. 4). We simulated mPSC recordings with frequencies of 1, 2, and 3 Hz. Despite the limits of detection, both event distributions and observed frequencies reflected the change in input, albeit strongly underestimating true frequencies (Fig. 4*D*). Mean mPSC amplitudes showed little change. Although these data confirm that pure mPSC frequency changes can be reliably followed, such effects cannot be concluded from 'real-world' data, as observed changes in mPSC

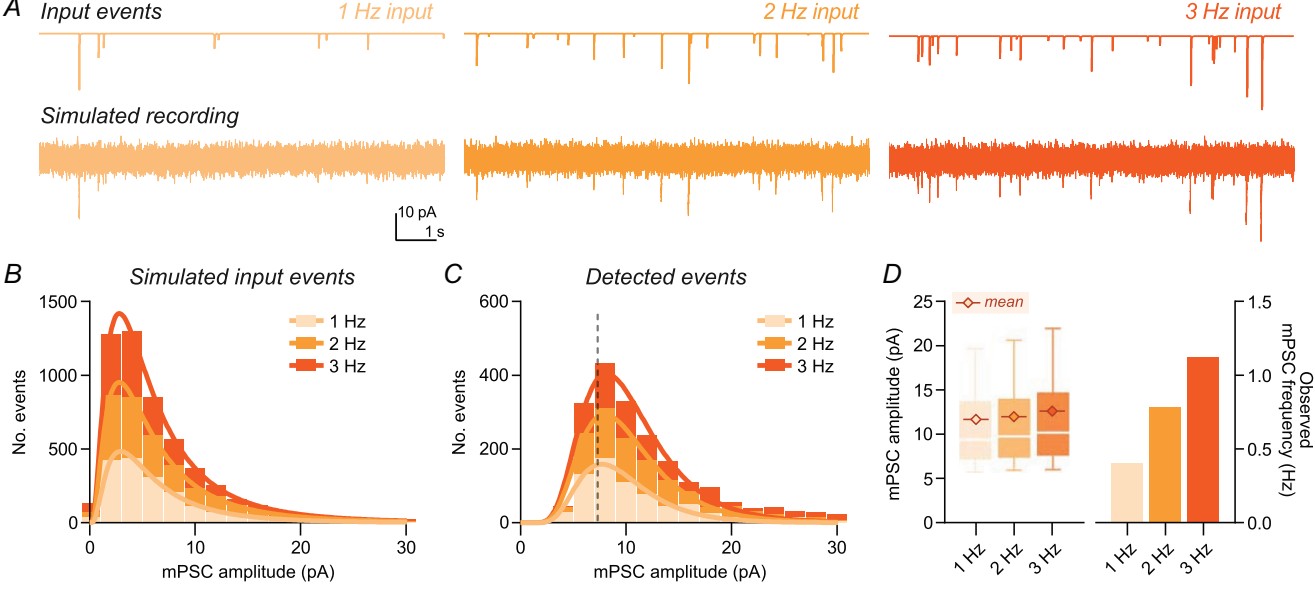

**Figure 4. Specific changes in event frequency are correctly represented as detected frequency changes**
*A*, example traces from simulated input at increasing miniature postsynaptic current (mPSC) frequency (1–3 Hz). *B*, input event histograms skewed purely in the *y*-axis with increasing frequency. *C*, detection of mPSC frequency changes was not distorted by detection. Histograms use a 2 pA bin width; continuous lines present lognormal fit of binned data, and the dashed line indicates calculated detection limit using the event scaling method (7.28 pA). *D*, input frequency change produced specific changes in detected frequency (right; '1 Hz', 0.4 Hz; '2 Hz', 0.8 Hz; '3 Hz', 1.1 Hz), and little change in detected amplitude (left, mean amplitude: '1 Hz', 11.7 pA; '2 Hz', 12.0 pA; '3 Hz', 12.6 pA).

frequency could occur through biological changes in either amplitude or frequency.

## Event kinetics influence detection

In addition to amplitude and frequency, event kinetics will vary between experimental and biological conditions. Changes in synaptic receptor properties, dendritic location of input events, and series resistance of patch-clamp recordings may all determine recorded event kinetics. We simulated recordings using event distributions with a range of amplitudes but fixed kinetics, slowing $\tau_{rise}$ and $\tau_{decay}$ in tandem between conditions (Fig. 5*A* and *B*). This manipulation mimics the kinetic filtering of either dendritic location or altered series resistance. Despite equivalent distributions of input event amplitudes (Fig. 5*C*), we observed large differences in detected event distributions (Fig. 5*D*). Smaller events, close to the detection limit, were detected better with fast than slow kinetics. The loss of small, slow events led to an increase in the mean detected event amplitude and a decrease in observed frequency for slower $\tau_{rise}$ and $\tau_{decay}$

conditions (Fig. 5*E*). Although the magnitude of this effect will be dependent on the event detection method employed, these results demonstrate that even pure differences in event kinetics can cause changes in detected event amplitude and frequency, further compromising biological interpretation.

## Real-world mPSC data contain hidden distributions

Although simulated data are valuable for testing mPSC analysis in a controlled system, we sought to assess the reliability of mPSC detection using experimentally recorded mPSCs. Many pharmacological manipulations are known to enhance synaptic transmission, but due to the complexity of this system it is difficult to confidently change transmission amplitude without frequency effects. However this can be achieved electronically. Recording mPSC datasets from the same neuron at two holding potentials allows specific and calculable modification of mPSC amplitude by increasing the driving force for ion flow across the membrane, in theory without influencing mPSC frequency. This approach has been pre-

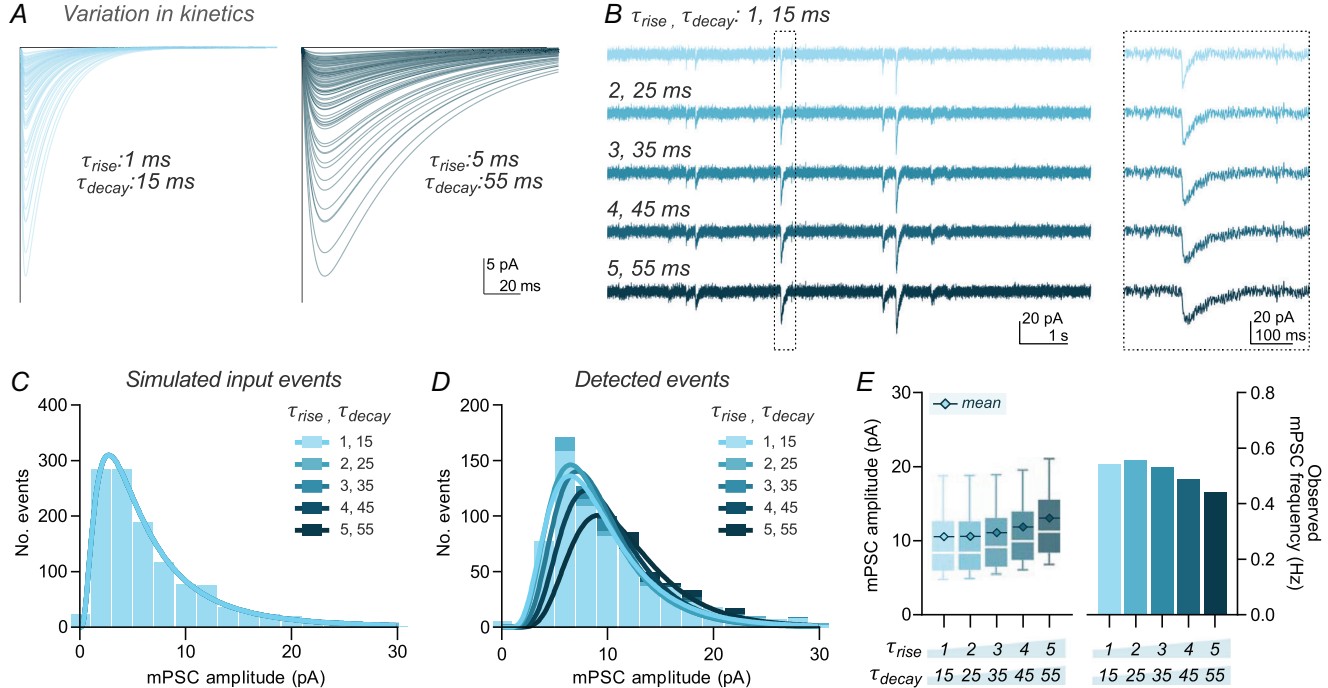

**Figure 5. Small events with slower kinetics are more likely to be lost in recording noise**
*A*, simulated miniature postsynaptic current (mPSC) events with equivalent amplitudes but differing rise and decay kinetics were generated. One hundred example events with fast (left) and slow (right) kinetics are depicted. *B*, events with five kinetic profiles were embedded in simulated recording noise for detection ($\tau_{rise}$ 1, 2, 3, 4 or 5 ms and $\tau_{decay}$ 15, 25, 35, 45 or 55 ms, respectively). Zoomed box depicts a single event across kinetic profiles. *C*, the distribution of input event amplitudes was no different across conditions. *D*, detected event histograms showed lower detection of small events for slower kinetic profiles. Histograms use 2 pA bin width and are overlaid with a lognormal fit (continuous line). *E*, the mean detected event amplitude increased with slower kinetic profiles (mean amplitude: 'rise 1': 10.6 pA, 'rise 2': 10.6 pA, 'rise 3': 11.1 pA, 'rise 4': 11.9 pA, 'rise 5': 13.1 pA), whereas the frequency of detected events was lower for slower kinetic profiles due to the loss of small events (detected frequency: 'rise 1': 0.54 Hz, 'rise 2': 0.56 Hz, 'rise 3': 0.53 Hz, 'rise 4': 0.49 Hz, 'rise 5': 0.44 Hz).

viously employed as an experimental control (Malgaroli & Tsien, 1992; Manabe et al., 1992) and to dissect synaptic parameters (Chen et al., 2015). We performed mEPSC recordings from CA1 pyramidal neurons in mouse organotypic slice cultures at both −70 mV and −90 mV holding potentials (−60 mV and −80 mV command potentials corrected for −10 mV liquid junction potential). These recordings were performed in the presence of 1 μM tetrodotoxin to block action potential generation, 10 μM SR-95 531 (Gabazine) to isolate mEPSCs and 100 μM D-APV to isolate the AMPAR current (Fig. 6*A*). Given the variability in real-world synaptic data, we sampled equal length recordings of mEPSCs at both holding potentials from every included cell and proceeded with mEPSC detection and analysis. The difference in holding potential should cause a 1.3-fold scaling in mEPSC amplitude. However, we observed no

change in mean mEPSC amplitude and instead observed an increase in mEPSC frequency (Fig. 6*C*). Event distributions showed *y*-axis scaling as expected from a frequency change, directly replicating our theoretical analysis (Fig. 6*B*). We confirmed that there was no difference in recording noise levels between conditions, ensuring our conclusions were not affected by detection variability (Fig. 6*D*). In addition, there was no substantial difference in series resistance between recording groups, which could have potentially altered the profile of recorded events, even in the absence of biological changes (mean ± SD of $R_s$: −70 mV, 12.9 ± 2.5 MΩ; −90 mV, 14.1 ± 2.9 MΩ). These observations corroborate our simulations, demonstrating that real-world mPSC analysis is confounded by the detection limit, and that a large number of mPSC events occur below the detection limit at hippocampal CA1 synapses.

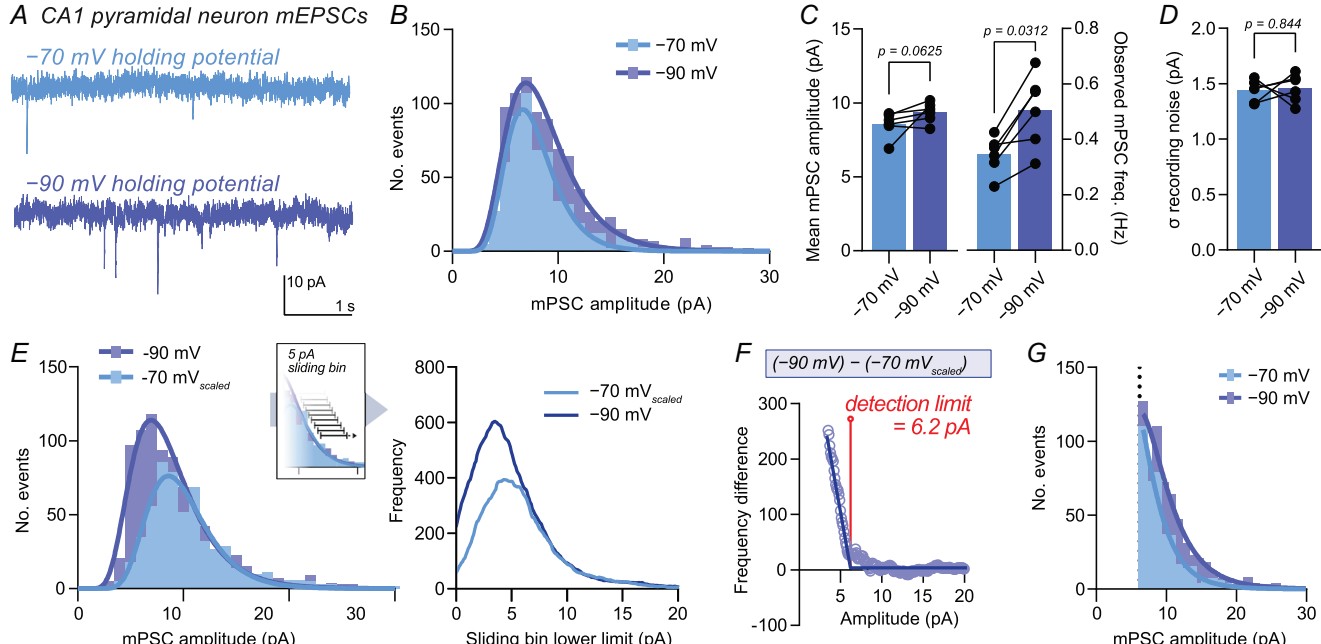

**Figure 6. Real-world mEPSC data show misrepresentative frequency increase in response to event amplitude manipulation**

*A*, example mEPSC traces recorded from CA1 pyramidal neurons at −70 and −90 mV holding potentials, where a pure event amplitude change is expected. *B*, histograms of recorded events suggested an overall frequency increase (lognormal fit, 1 pA bin width). *C*, observed mean event amplitudes were not affected by holding potential-driven amplitude increase (mean ± SD: −70 mV, 8.54 ± 0.86 pA; −90 mV, 9.37 ± 0.68 pA; *n* = 6 cells; Wilcoxon matched-pairs signed-rank test, *P* = 0.0625), yet observed frequency was increased by 50% (mean ± SD: −70 mV, 0.34 ± 0.07 Hz; −90 mV, 0.51 ± 0.13 Hz; *n* = 6 cells; Wilcoxon matched-pairs signed-rank test, *P* = 0.0312). *D*, there was no difference in the SD (σ) of recording noise between conditions (mean ± SD: −70 mV, 1.44 ± 0.10 pA; −90 mV, 1.46 ± 0.13 pA; *n* = 6 cells; Wilcoxon matched-pairs signed-rank test, *P* = 0.844). *E*, scaling the −70 mV dataset by 1.3 produced overlapping distributions at high amplitudes. These datasets were resampled by a 5 pA-sliding bin at a resolution of 0.1 pA (right). *F*, plotting the difference between resampled datasets allowed the determination of the amplitude at which events begin to be 'lost' at −70 mV but not at −90 mV. The frequency difference curve was fitted with a broken stick relationship, which approximated the detection limit (6.2 pA or 3.9σ of highest recording noise (highest noise: 1.61 pA)). *G*, distribution of recorded mEPSCs after application of the detection limit as a cut-off. Histogram bins start from the detection limit, with 1 pA bin width and a lognormal fit (continuous line). No peak to event distribution was observed in the recorded range; therefore the modal event amplitude lies beneath the detection limit.

## Experimental estimation of the detection limit

We have demonstrated the fallibility of mPSC analysis for understanding and interpreting biological effects. We next sought to determine a means to improve the reliability of mPSC analysis through determination of the event detection limit. Using holding potential scaling we were able to 'visualise' mPSC events that exist but were hidden beneath the detection limit when recording at −70 mV. mEPSC distributions recorded and detected using a −70 mV holding potential were scaled to their expected amplitude at −90 mV (×1.3) and plotted alongside our 'ground truth' −90 mV mEPSC dataset (Fig. 6E). Although these distributions overlay almost perfectly at high event amplitudes (away from the detection limit), the −70 mV$_{scaled}$ dataset showed lower frequency of small amplitude events. Our −90 mV dataset was also likely to be incomplete at small amplitudes, yet contained a more complete representation than that recorded at −70 mV. Therefore the point at which these two curves diverge is the amplitude at which mPSCs begin to become undetected, i.e. the detection limit. A sliding bin histogram of mPSC events was plotted for −90 mV and scaled datasets (Fig. 6E). Calculating the difference between these curves (subtraction of −70 mV$_{scaled}$ from −90 mV) produces a biphasic curve, where high amplitudes can be fit with a $y = 0$ curve (no difference in event detection), yet low amplitudes follow a linear relationship with negative gradient (Fig. 6F). This graph can be approximated with a 'broken stick' curve, where the break point represents the lowest amplitude at which zero false negatives are recorded: the 'detection limit' (Fig. 6F). In our dataset the detection limit was estimated at 6.2 pA or 3.9 times the highest standard deviation ($\sigma$) of recording noise across all recordings (1.61 pA).

Finally we used this knowledge to re-examine recorded event distributions. We applied the estimated detection limit as a cut-off for included events and replotted recorded datasets with the minimum bin value beginning at this cut-off (Fig. 6G). We were unable to fit a peak to our real-world mEPSC dataset, demonstrating that the modal event remains below the detection limit. Estimates of quantal size or in-depth interpretation of distribution changes would therefore not be possible from such data.

## Discussion

The properties of central synapses are highly diverse, both across the brain and at the level of individual neurons. For this reason, understanding the changes in synaptic properties underlying brain function requires robust methods for their study. Analysis of mPSCs has

the potential to provide information about heterogeneous synaptic efficacies across the neuronal dendritic tree. This approach is powerful and technically simple, but the pitfalls of data analysis and interpretation are deep, hidden, and currently not so widely appreciated. We have demonstrated these issues using both simulated and experimental datasets, suggesting analyses for careful interpretation of recorded data. Although this manuscript focuses on mini analysis (mPSCs), these concepts are directly applicable to sPSC recordings or analysis of any detected event that has close proximity to noise levels.

## Empirical interpretation of mPSC datasets

It would be logical to assume that measured changes in mean event frequency represent underlying changes in mean event frequency and, similarly, event amplitudes. However our data reiterate that this is a false assumption. When events are embedded in recording noise, changes in mPSC amplitude are more robustly detected as changes in frequency than mean event amplitude (Figs 3 and 6). Therefore, even specific changes in observed mPSC frequency could be caused by underlying changes in either frequency or amplitude. This interdependence was well appreciated in the early years of mini analysis, with frequent references to this phenomenon in data interpretation (Diamond & Jahr, 1995; Manabe et al., 1992; Mennerick & Zorumski, 1995; Yamada & Tang, 1993). Although this knowledge still exists with the more physiologically minded, widespread appreciation of the effect often appears to be forgotten.

The effect of incomplete detection demonstrates the importance of interpreting event distributions rather than average values; however distributions can be similarly misleading. If complete distributions are seen above the noise level, interpreting mPSC changes is not an issue, and even mean values will in some way reflect underlying biological changes. However for the majority of brain synapses this is not the case. The most intensively studied excitatory connections across the hippocampus and cortex are weak and have many events buried in noise. In our hands, the modal mEPSC of CA1 pyramidal neurons lies beneath recording noise (<6.2 pA at −70 mV holding potential).

Unfortunately the detection limit is not simply a sharp cut-off. Small events are lost with increasing likelihood the smaller they are, creating a false 'peak' to event amplitude distributions. The resulting profiles strongly resemble the lognormal-like shape expected from synaptic events, giving false confidence that measured data fully represent the underlying biology. Quantal analysis from miniature or spontaneous PSC data is highly problematic unless

either clear evidence of quantal properties is observed (Paulsen & Heggelund, 1994), or the peak of the event distribution can be unequivocally observed above the detection limit. Even then, variability between individual synapses, smearing of distributions by dendritic filtering, and space-clamp problems will almost certainly preclude simple interpretation of such results (Edwards et al., 1976b; Jack & Redman, 1971; Malinow, 1991; Redman, 1990; Williams & Mitchell, 2008).

It is not only event amplitude changes that can be difficult to interpret. Changes in event kinetics can occur biologically through either differences in synaptic receptor composition (Greger et al., 2017; Jonas & Spruston, 1994) or aforementioned changes in input location on the dendritic tree (Jack & Redman, 1971; Rall et al., 1967), while also being sensitive to recording conditions such as series resistance. Event kinetics also affect detection, with small slow events less likely to be detected in our analysis. Kinetic changes can therefore also misrepresent mini analysis results, potentially appearing as changes to synaptic event amplitudes or frequencies in post-detection distributions. Evoked synaptic responses comprise multiple small release events at variable and possibly distributed dendritic locations; therefore considerations of input location and dendritic filtering also have the potential to misrepresent synaptic changes interpreted from this method.

To facilitate data analysis, we present a means to estimate the detection limit. Using different voltage clamp holding potentials, electronically scaled mPSCs can be compared to determine the amplitude at which events reliably emerge from noise (Fig. 6). Space-clamp problems will prevent this from being a perfect measure (Williams & Mitchell, 2008), but the approach provides an experimental estimate of the amplitude at which mPSC analysis becomes unreliable for individual recording set-ups and configurations. Event distributions can then be attenuated so as to analyse only the events falling above this point, preventing false peaks in amplitude distributions. It is important to note that the detection limit is not a strict value, as seen from the probabilistic loss of small events in our analysis (see also (Clements & Bekkers, 1997)). For practical reasons however, we define the detection limit as the point at which events begin to be lost. Our detection limit estimation from real-world data is in line with previous estimates; Clements and Bekkers predicted that $4 \times \sigma$ would eliminate false-negative detection (Clements & Bekkers, 1997). Therefore where empirical detection limit measurement is not possible, $4\sigma$ of the noisiest included recording may be an appropriate cut-off for mPSC analysis. Critically, when binning data attenuated at a calculated detection limit (e.g. histogram presentation), lower bin limits must start precisely at the event cut-off, most likely requiring non-integer edge values.

## Practical steps for minimising misinterpretation due to recording conditions

Not only are mPSC frequency changes an unreliable indicator of biological changes, but changes in mean amplitude are more sensitive to the level of recording noise than to actual underlying synaptic changes. Without careful analysis of recorded noise levels between conditions, misinterpretation of synaptic changes from mini analysis is very likely. Recording noise is dependent not only on set-ups but also on individual cells. The quality of sealing and membrane integrity during whole-cell recordings will influence the standard deviation of noise, and event detection will be affected in turn. It is important that analysed mPSC recordings have as low noise as practically achievable. More important still is that experimental groups have equivalent noise levels between conditions to prevent differences in detection from artificially resulting in detected changes. This likely means that not all recorded cells will be included in final analyses, and that higher noise recordings will need to be discarded to ensure comparability between conditions. When experimental groups have different levels of noise, applying the detection threshold from the highest noise dataset to all conditions may limit errors due to differences in detection. To ensure maximal validity of experimental conclusions it would be 'best practice' for the standard deviation of noise between conditions to be presented alongside any mini analysis dataset.

Series resistance is the second recording parameter that can influence mini analysis results. Patch-clamp recordings with higher series resistance will filter synaptic events to have smaller recorded amplitudes and slower kinetics (Armstrong & Gilly, 1992; Barbour, 2018). High series resistance will also affect the quality of voltage clamp, which, in turn, may compromise recorded data. Therefore as with recording noise, it is also important both to have as low series resistance recordings as practical, and to ensure that the series resistance of recordings is similar between compared datasets. Finally, temperature has a strong influence on the properties of synaptic transmission (Hardingham & Larkman, 1998; Thompson et al., 1985). This too should be controlled and consistent between recorded conditions.

We summarise these considerations in a nine-point plan for improved mini analysis:

Before/during recording:

1 **Ensure recording set-up is optimised with minimal noise.**
2 **Assess series resistance continuously during data acquisition.** Typically with a 2–10 mV hyperpolarising 'test pulse' approximately every 30–60 s. This can also be used to monitor input resistance and cell capacitance for measures of cell properties and recording quality.

Before/during data analysis:

3 **Discard any recordings with poor series resistance or where series resistance changes substantially during recordings.** Typically cut-offs of 20 MΩ and a change of <20% are appropriate limits to balance quality and achievability.

4 **Manually inspect all recordings for periods of instability.** These areas are likely to have higher recording noise and the potential to introduce false-positive events. These periods should be excluded.

5 **Measure and present the standard deviation of baseline noise for each recording and compare between conditions.** Ensure that the measured region/s are free from visible mPSC events. Assess whether conditions have equivalent noise levels before subsequent analysis. Discard recordings with high or non-representative noise levels that would skew comparisons.

Note: it should not be unusual to exclude a large fraction of recorded cells where necessary to maintain high data quality in subsequent analyses.

6 **Determine the number of cells and analyse recording duration for each condition.** Analyse equivalent recording duration for every cell so that datasets are not skewed towards cells with longer recording duration. Consider also plotting amplitude distributions with equivalent *number of events* per cell, so that data are not skewed to cells with higher mPSC frequency.

7 **Detect events and apply detection limit cut-off.** Ideally the detection limit is estimated experimentally using the event scaling method detailed here or, alternatively, by $4\sigma$ of the noisiest included recording.

It is also recommended to visualise detected events to confirm valid detection.

8 **Plot event distributions starting from the detection limit.** Use the cut-off selected above as the lower limit of the first histogram bin to prevent an incomplete first bin from creating an artificial peak in the resulting distribution.

9 **Interpret with care.** Remember that many biological and experimental factors give rise to the resulting distributions.

Further practical considerations for recording spontaneous events are discussed in Hartveit and Veruki (2007).

## Biological interpretation of mPSC changes

Interpretation of 'mini data' extends beyond just numbers and distributions. As we understand more about synaptic function it becomes clear that the historic doctrine of 'mPSC amplitude changes = postsynaptic' and 'mPSC frequency changes = presynaptic' is an oversimplification. Multiple biological factors of both pre- and postsynapse can alter both mPSC frequency and amplitude (Fig. 7). Changes in mPSC amplitude could be caused postsynaptically by changes in neurotransmitter receptor abundance or conductance (Kessels & Malinow, 2009; Malenka & Nicoll, 1999), but also presynaptically or transsynaptically, through vesicle properties or alignment (Scheefhals & MacGillavry, 2018; Shi et al., 2022). Similarly, changes in the number of active release sites could change mPSC frequency (Malgaroli & Tsien, 1992), but postsynaptic unsilencing

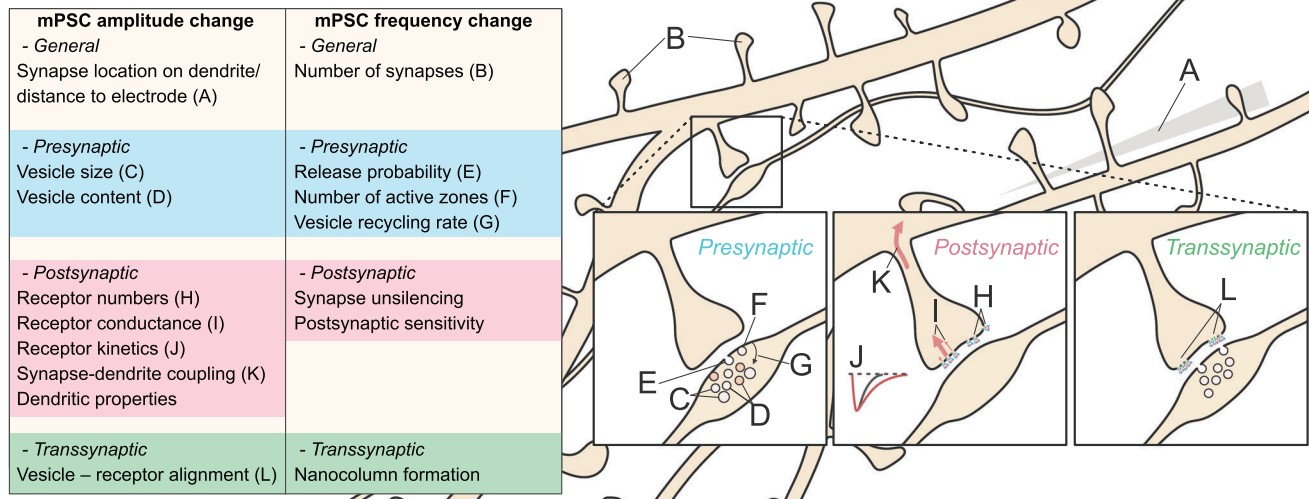

| mPSC amplitude change | mPSC frequency change |
|---|---|
| *- General* | *- General* |
| Synapse location on dendrite/ distance to electrode (A) | Number of synapses (B) |
| *- Presynaptic* | *- Presynaptic* |
| Vesicle size (C) | Release probability (E) |
| Vesicle content (D) | Number of active zones (F) |
| | Vesicle recycling rate (G) |
| *- Postsynaptic* | *- Postsynaptic* |
| Receptor numbers (H) | Synapse unsilencing |
| Receptor conductance (I) | Postsynaptic sensitivity |
| Receptor kinetics (J) | |
| Synapse-dendrite coupling (K) | |
| Dendritic properties | |
| *- Transsynaptic* | *- Transsynaptic* |
| Vesicle – receptor alignment (L) | Nanocolumn formation |

**Figure 7. Overview of possible factors influencing mPSC changes**
Both miniature postsynaptic current (mPSC) frequency and amplitude may be altered by a range of pre (blue), post (red) and transsynaptic (green) changes, in addition to more general properties (beige), complicating interpretation of recorded observations. The factors presented here do not include differences in recording configuration, which will also affect the conclusions drawn from final datasets.

(Isaac et al., 1995; Liao et al., 1995) could enact a post-synaptic change in event frequency. Neuronal properties, such as dendritic event location and cable properties, will also influence mPSC parameters (Jack & Redman, 1971; Rall et al., 1967), and should large changes in cell parameters such as cell capacitance or membrane resistance occur between compared conditions, their effect on recording conditions should be considered. Each of these factors can add significant complexity to the distribution of synaptic events recorded from across a neuron's dendritic tree. Therefore interpreting specific effects, in particular detailed changes such as effects on synaptic nanoarchitecture, would need both low noise recordings and particularly careful data analysis.

Although minis are a simple means to acquire functional synaptic data, it is important to note that the synaptic mechanisms for spontaneous vesicle release and action potential-dependent synaptic transmission appear to be distinct (Peled et al., 2014; Sara et al., 2005). Therefore, compounded by the issues with data interpretation highlighted above, translating mini data into a mechanistic understanding of functioning brain circuits will most likely require complementary investigations of synaptic properties to be performed in parallel.

'Mini analysis' has become widespread due to the ease of single-cell recording to acquire synaptic insights. Although powerful, simple, and widely employed, this approach is prone to misinterpretation. We hope that this study can aid robust data analysis, strengthening insightful synaptic physiology and neuroscience research.

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

## Additional information

### Data availability statement

Mini detection software from the BioSig project was used, with code available at https://biosig.sourceforge.net/. All additional code is publicly available at https://github.com/jakefwatson/miniplace.

### Competing interests

The authors have no competing interests to declare.

### Author contributions

J.F.W. conceived and conducted the project and wrote the manuscript. I.H.G. acquired funding and revised the manuscript draft. Both authors approved the final version of the manuscript and agreed to be accountable for all aspects of the work in ensuring that questions related to the accuracy or integrity of any part of the work are appropriately investigated and resolved. Both authors qualify for authorship, and all those who qualify for authorship are listed.

### Funding

Funding was provided by the Medical Research Council (MRC – MC_U105174197 to I.H.G.) and the European Union's Horizon 2020 programme through a Marie Skłodowska-Curie Actions Individual Fellowship (MSCA-IF 101026635 to J.F.W.).

### Acknowledgements

This work was supported by Biological Services teams at both the Laboratory of Molecular Biology and Ares facilities. The authors are very grateful to Prof. Helmut Kessels and Dr. Hinze Ho for initial discussions that led to this study, Dr. Andrew Penn for constructive feedback on the project, Xinyao Dou for comments on the study, and Profs. Peter Jonas and Roger Nicoll for feedback on the manuscript.

### Keywords

detection limit, event detection, mEPSCs, mini analysis, mPSCs, noise, patch-clamp, synapse, synaptic transmission

### Supporting information

Additional supporting information can be found online in the Supporting Information section at the end of the HTML view of the article. Supporting information files available:

**Peer Review History**

