## [Peer Review History · The Journal of Physiology]

‘Mini analysis’ misrepresents changes in synaptic properties due to incomplete event detection

Ingo Harald Greger and Jake F Watson

DOI: 10.1113/JP288183

Corresponding author(s): Jake Watson (jake.watson@ist.ac.at)

The following individual(s) involved in review of this submission have agreed to reveal their identity: Ian D. Forsythe (Referee #2)

Review Timeline:

Submission Date:	20-Nov-2024
Editorial Decision:	02-Jan-2025
Revision Received:	04-Aug-2025
Editorial Decision:	26-Aug-2025
Revision Received:	03-Sep-2025
Accepted:	08-Sep-2025

Senior Editor: Katalin Toth

Reviewing Editor: Conny Kopp-Scheinflug

Transaction Report:

Dear Dr Watson,

Re: JP-RP-2024-288183 "'Mini analysis' is an unreliable reporter of synaptic changes" by Ingo Harald Greger and Jake F Watson

Thank you for submitting your manuscript to The Journal of Physiology. It has been assessed by a Reviewing Editor and by 2 expert referees and we are pleased to tell you that it is potentially acceptable for publication following satisfactory major revision.

REVISION CHECKLIST:

Please upload two versions of your manuscript text: one with all relevant changes highlighted and one clean version with no

changes tracked. The manuscript file should include all tables and figure legends, but each figure/graph should be uploaded as separate, high-resolution files.

We look forward to receiving your revised submission.

Yours sincerely,

Katalin Toth
Senior Editor
The Journal of Physiology

REQUIRED ITEMS

- Author photo and profile. First or joint first authors are asked to provide a short biography (no more than 100 words for one author or 150 words in total for joint first authors) and a portrait photograph. These should be uploaded and clearly labelled together in a Word document with the revised version of the manuscript. See Information for Authors for further details.
- Please upload separate high-quality figure files via the submission form.
- Please ensure that the Article File you upload is a Word file.
- Please include an Abstract Figure file, as well as the Figure Legend text within the main article file. The Abstract Figure is a piece of artwork designed to give readers an immediate understanding of the research and should summarise the main conclusions. If possible, the image should be easily 'readable' from left to right or top to bottom. It should show the physiological relevance of the manuscript so readers can assess the importance and content of its findings. Abstract Figures should not merely recapitulate other figures in the manuscript. Please try to keep the diagram as simple as possible and without superfluous information that may distract from the main conclusion(s). Abstract Figures must be provided by authors no later than the revised manuscript stage and should be uploaded as a separate file during online submission labelled as File Type 'Abstract Figure'. Please also ensure that you include the figure legend in the main article file. All Abstract Figures should be created using BioRender. Authors should use The Journal's premium BioRender account to export high-resolution images. Details on how to use and access the premium account are included as part of this email.
- Please include a full title page as part of your main article (Word) file, which should contain the following: title, authors, affiliations, corresponding author name and contact details, keywords, and running title.
- Author photo and profile. First or joint first authors are asked to provide a short biography (no more than 100 words for one author or 150 words in total for joint first authors) and a portrait photograph. These should be uploaded and clearly labelled together in a Word document with the revised version of the manuscript. See Information for Authors for further details.

- Please upload separate high-quality figure files via the submission form.

- Please ensure that the Article File you upload is a Word file.

- Please include an Abstract Figure file, as well as the Figure Legend text within the main article file. The Abstract Figure is a piece of artwork designed to give readers an immediate understanding of the research and should summarise the main conclusions. If possible, the image should be easily 'readable' from left to right or top to bottom. It should show the physiological relevance of the manuscript so readers can assess the importance and content of its findings. Abstract Figures should not merely recapitulate other figures in the manuscript. Please try to keep the diagram as simple as possible and without superfluous information that may distract from the main conclusion(s). Abstract Figures must be provided by authors no later than the revised manuscript stage and should be uploaded as a separate file during online submission labelled as File Type 'Abstract Figure'. Please also ensure that you include the figure legend in the main article file. All Abstract Figures should be created using BioRender. Authors should use The Journal's premium BioRender account to export high-resolution images. Details on how to use and access the premium account are included as part of this email.

- Please include a full title page as part of your main article (Word) file, which should contain the following: title, authors, affiliations, corresponding author name and contact details, keywords, and running title.

Reviewing Editor's comments:

Your manuscript has been reviewed by two experts in the field and both arrived at the same conclusion: It is essential to remind researchers about the limitations of their methods. However, the manuscript downplays how widely recognized these issues have been in the literature but appropriately stresses the importance of bringing them to the attention of a new generation of neuroscientists, particularly those who may not fully understand the limitations of single-electrode whole-cell voltage clamp methods. So I strongly agree with both reviewers, that your manuscript will be a valuable publication to enforce high-quality electrophysiology, but you should step away from the sensationalist title and wording. I recommend a serious revision according to the reviewers' detailed suggestion before resubmission. Upon resubmission, please also report the origin of the animals, their access to food and water as well as the terminal procedures.

Referee #1:

In this manuscript, the authors perform a detailed signal-to-noise assessment of spontaneous miniature events recorded using whole cell patch clamp electrophysiology. Their analysis shows that a large fraction of spontaneous events are missed or detected incompletely leading to inaccuracies regarding synaptic changes. None of these findings are surprising and they

are all well-known to those who analyze spontaneous events with rigor and make interpretations with caution. In addition, the arguments would apply to any measurement where signals need to be discerned from background noise including optical measures of single synapse responses. While this manuscript could be a valuable contribution by reminding the field things that may have been overlooked (or "forgotten"), its rather "sensationalist" narrative, as exemplified by the title "'Mini analysis' is an unreliable reporter of synaptic changes" distracts from a potentially balanced and level-headed message.

1. The authors should change their attention-grabbing title to something more relevant to the contents of the paper regarding signal to noise assessments for miniature synaptic event analysis. The Biorxiv version of this article has been downloaded several times and therefore its title already served the purpose.
2. The analysis does not consider alterations in kinetics of events, especially rise times. This is typically an important criterion in event selection (as in the case of author's protocol). However, changes in this detection criteria may result in dramatic changes in interpretation.
3. The authors do not seem to consider dendritic filtering in their assessment. This biological factor also disadvantages detection of events far from the recording electrodes (and distorts their size and kinetics). This factor is a likely contributor to the small events buried in the noise and as such distort analyses.
4. Along the same lines, how do these factors impact detection of evoked events? As they represent summation of several miniature events from across a dendritic tree, small events originating from distal sites will likely be misrepresented in potential synaptic changes.
5. The authors also do not consider the role of several recording parameters. For instance, cell capacitance, series resistance, membrane resistance as well as multi-compartment morphology of central neurons would all be factors complicating voltage/space claim accuracy and noise.
6. In addition, a fundamental reason why mini frequency may not be a good indicator of synaptic changes is that when assessed at the single synapse level, it often does not correlate with evoked release probability.

Referee #2:

Overall. This article provides some important insights into the limitations of estimates of synaptic parameters from measurement of miniature PSC events. The introduction underestimates how well known many of these issues are (were) in the literature but correctly recognises the importance of raising these issues to another generation of neuroscientists; particularly those that do not fully appreciate the limitations of the single electrode whole-cell voltage clamp. The background and appreciation of these limitations is much greater than these authors seem to acknowledge in their introduction, but perhaps this also reflects the general awareness of synaptic experimentalists of the limitations of their technique; so the focus here is timely and will help correct/improve synaptic studies for the future. It is important that the authors conduct a thorough re-write of the whole text to reduce ambiguity and make their conclusions clear to the reader (see figure legend summary sentences for instance).

My specific points (minor and major) are listed by line number

Abstract

20. Just say extensively, delete 'one of the most', and.

21. Delete 'approaches'.

Re-write the abstract - the first half is introduction - your abstract should focus on why you have conducted the study, what you have done and provide more specific details of your results. Many of the sentences contain empty phrases.

31/32. - Sentence is confusing or poorly constructed..... Measured changes in mIPSC amplitude or frequency, falsely report changes in the synaptic parameters, due to incomplete detection of the distribution and magnitude of the synaptic events....?

34. Delete 'not only'

35. Delete 'but also' > and

Introduction

The introduction needs an extensive rewrite; it currently fails to do justice to the long history of quantal analysis and does not provide a broad introduction to the subject. The last paragraph needs to make clear what data used here is *in silico* and which is animal experimental.

42. Delete the first sentence, and the second or find a better means of expression. We cannot know that each of trillions of synapses has unique properties!

47. Your introduction to quantal synaptic events should be broader and can include the neuromuscular junction. Your citation in the introduction sometimes lacks depth, and includes too many reviews over original sources (e.g. Kavalali 2015; Edwards 1991).

64. Expand to provide information or delete this sentence. The remainder of the sentence makes statements without citation, but perhaps you will deal with these matters later in your manuscript. It would be better to cite, as your bald statements do not help the uninitiated to access your article. You do not need to cite every single thing, but there should be sufficient to satisfy the knowledgeable reader that you know the historical context whilst also providing background for those seeking an introduction to the basics.

Citation:

It is surprising that you do not take a broader view of quantal synaptic physiology, nor cite or discuss the work of Redman, Edwards and Walmsley in the late 1970's and 80s. While you are working on central synapses, many of the fundamentals (and limitations) were studied at the NMJ. There are other studies of a detection limit and discussions of technical limitations. I see that you acknowledge this in the discussion, but you need not build 'straw men' for an important paper.

Methods:

These are normally placed after the introduction.

Why were experiments conducted at room temperature, rather than Physiological temperatures? Perhaps a note about the impact of temperature on fluctuations and noise in real recordings might be appropriate.

Why did you use Cs methylsulphonate as your internal ion? Are you aware that the very low internal chloride ion concentration would have generated huge junction potentials? You should state what those junction potentials were, even if you have not corrected for them. Low chloride internals can also cause problems in current passing, as silver chloride is the interface of your silver wire with the electrolyte....

What was the mean and range of series resistances of your recording pipettes? These should not be those data measured at the start of the recording, but measured at the time of collecting a given data set (as you know series resistance often increases rapidly following the initial break-in perhaps due to plugging of the pipette lumen).

All abbreviations need to be specified and drugs used named in full, and their source (at least in the methods).

You need some statements about statistics.

You must review and fulfil the detailed information stated in the instructions to authors about use of Animals in Research and include the requested information in your methods section. Some of the information is stated but not all.

Figures

All figures and graph error bars need SEM to be replaced with Standard Deviation (SD).

The first sentence of each legend should ideally give the take-home message for that figure. This nearly works for Fig 1 - but could be clearer: e.g. "The limit to detection misrepresents the event distribution by failing to include small events below the detection limit." The other figure legends need something similar.

Figure 1

1C Please add your calculated detection limit to the graphs, perhaps with a vertical dashed line?

1D, 1G - Ambiguous axis titles - please change to: Mean mPSC Amp. (pA); Observed mPSC Fq. (Hz)

Figure 2

2C, 2D and 2G. Please add your calculated detection limit to the graphs, perhaps with a vertical dashed line?

Figure 3

3D - Ambiguous axis titles - please change to: Mean mPSC Amp. (pA); Observed mPSC Fq. (Hz)

Figure 4.

4C - Ambiguous axis titles - please change to: Mean mPSC Amp. (pA); Observed mPSC Fq. (Hz)

Legend. You might measure or determine the detection limit.... You do not detect the detection limit.

Figure 5 Diagram.

This contains some useful information in the table but there are issues such as vesicle recycling and the interaction of voltage-clamp with synaptic location which are omitted.

Results

92. This phrase reads well but is not very scientific 'establishing a pipeline for simulating mPSC recordings', please state that simulated mPSCs were generated as detailed in the methods.

121. 'by at least a factor of two'.

186. to assess the unreliability > to assess the reliability. Seems like a less biased statement?

195. Using driving force to change synaptic amplitude has at least two important pitfalls under voltage-clamp conditions. The fact that these issues may have caused an underestimation of the holding potential is implied by the relatively small measured changes in the mPSC amplitude (Fig 4B; looking only at those events larger than the detection limit of 5.9pA). As you state in your text, I would have expected an increase of between 25-33% on changing the driving force from -60 to -80 mV (assuming a reversal potential of 0mV for an excitatory glutamatergic synapse). Your measured mPSCs increase by much less than this (more like 10-15%) in 4B. While you take this to support your hypothesis, it is as likely that you have not achieved your command voltage change (from -60 to -80 mV) at the active synapses due to (1) voltage drop across the patch pipette series resistance (2) decay of your command voltage due to cable theory and the neuron length constant (i.e. that distance between your pipette at the soma and the synapse on the dendrites). While your use of internal Cs will have raised your length constant - to your advantage; your use of low chloride has introduced a large uncompensated junction potential; so you never were at -60mV holding potential to start with.....This could further undermine the accuracy of your

voltage command. So you really need to know the value of your Junction potential. It would be particularly interesting if you were to apply your methods to data collected from preparations in which synapses are made only within a highly confined/proximal location to the recording site - for instance in an 'old fashioned' dissociated hippocampal preparation (where synaptic boutons remain attached to the soma and spontaneously release minis) or in a preparation such as the calyx of Held where most excitatory synapses form on the soma.

211. Fallibility. I would suggest that at least part of the overall problem is the quality of voltage-clamp and of course this is as big a problem for you as it is for other investigators. It need not be a terminal problem, as you could incorporate this aspect of the solution into your manuscript with a re-write. Indeed, I would also suggest that the value of your manuscript would be increased if you were to present methods by which you could estimate, compensate or place limits on the quality of experimental data.

427. In the past the Journal of Physiology requested confirmation that those acknowledged as having provided feedback on a manuscript had given permission for their names to be used.

END OF COMMENTS

JP-RP-2024-288183 Point-by-point response to reviewers

Reviewing Editor's comments:

Your manuscript has been reviewed by two experts in the field and both arrived at the same conclusion: It is essential to remind researchers about the limitations of their methods. However, the manuscript downplays how widely recognized these issues have been in the literature but appropriately stresses the importance of bringing them to the attention of a new generation of neuroscientists, particularly those who may not fully understand the limitations of single-electrode whole-cell voltage clamp methods. So I strongly agree with both reviewers, that your manuscript will be a valuable publication to enforce high-quality electrophysiology, but you should step away from the sensationalist title and wording. I recommend a serious revision according to the reviewers' detailed suggestion before resubmission. Upon resubmission, please also report the origin of the animals, their access to food and water as well as the terminal procedures.

We thank the reviewing editor and reviewers for their positive feedback on our work. We are also very grateful for the highly detailed and constructive comments from both reviewers. We have made substantial changes to the manuscript based on these comments, which have undoubtedly improved this paper. In particular, we have changed the title and substantially expanded the introduction to provide further background on both quantal analysis and how widely recognised these issues were in the literature.

We have now included an '*Animals*' subsection to the Materials and Methods to include the required information. Apologies for this oversight.

Please note: line numbers quoted below refer to those on the Merged PDF. These differ slightly to the provided Word document due to changes in the PDF building process.

Referee #1:

In this manuscript, the authors perform a detailed signal-to-noise assessment of spontaneous miniature events recorded using whole cell patch clamp electrophysiology. Their analysis shows that a large fraction of spontaneous events are missed or detected incompletely leading to inaccuracies regarding synaptic changes. None of these findings are surprising and they are all well-known to those who analyze spontaneous events with rigor and make interpretations with caution. In addition, the arguments would apply to any measurement where signals need to be discerned from background noise including optical measures of single synapse responses. While this manuscript could be a valuable contribution by reminding the field things that may have been overlooked (or "forgotten"), its rather "sensationalist" narrative, as exemplified by the title "'Mini analysis' is an unreliable reporter of synaptic changes" distracts from a potentially balanced and level-headed message.

We thank the reviewer for their view of our work as a 'valuable contribution', and also for the insightful comments that have helped us to make this manuscript into a more comprehensive resource for considering mini analysis issues.

1. The authors should change their attention-grabbing title to something more relevant to the contents of the paper regarding signal to noise assessments for miniature synaptic event analysis. The Biorxiv version of this article has been downloaded several times and therefore its title already served the purpose.

We had not intended the title of this manuscript to be provocative or attention-grabbing, and meant it more as a concise reflection of the conclusions of our paper. We appreciate the reviewer's feedback and have revised the title to more directly reflect the analyses we perform. We have also now emphasised that the limitations of incomplete detection were well recognised in the early days of mini analysis in the key points (line 21), abstract (line 34), and introduction (line 123) of the revised manuscript, in addition to the discussion as in the previous version. We hope the new title and background information more constructively conveys the results of our analyses.

2. The analysis does not consider alterations in kinetics of events, especially rise times. This is typically an important criterion in event selection (as in the case of author's protocol). However, changes in this detection criteria may result in dramatic changes in interpretation.

This is an important point, and we are grateful to the reviewer for raising it. We have now performed additional analysis varying event kinetics. We compared events with fast rise and decay (1 ms τ rise, 15 ms τ decay) to increasingly slower events. Small events with slow kinetics were more frequently undetected, resulting in higher detected mean amplitude, and lower detected event frequency. Also relevant for the reviewer's later points, these data highlight 1. that synaptic manipulations affecting event kinetics may also induce misrepresentative changes in mini analysis results, 2. manipulations affecting the dendritic location of mPSC input will alter event kinetics and misrepresent conclusions, and 3. how recording conditions e.g. series resistance can affect conclusions. This data is included in the new Figure 5 of the revised manuscript (line 348 of the results), and consideration of these points has been included in the revised discussion (paragraphs starting on line 465 and 509).

3. The authors do not seem to consider dendritic filtering in their assessment. This biological factor also disadvantages detection of events far from the recording electrodes (and distorts their size and kinetics). This factor is a likely contributor to the small events buried in the noise and as such distort analyses.

This is an important point. We had previously stated the impact of dendritic filtering in the discussion section, but this should have been made much clearer. We have now stated this effect as a major issue with mini analysis in relation to quantal analysis in the revised introduction (line 103), in discussion of data interpretation (lines 461, 467, 568), and have included dendritic location more prominently on the summary figure (Figure 7 of the revised manuscript).

4. Along the same lines, how do these factors impact detection of evoked events? As they represent summation of several miniature events from across a dendritic tree, small events originating from distal sites will likely be misrepresented in potential synaptic changes.

We agree that differential input location will also be a factor when studying evoked events. The degree of influence will highly depend on the method of event evocation, and the cell in question. With a typical 'minimal stimulation' it is likely that the evoked event is formed by

summation of spatially clustered synapses due to the spatial restricted nature of stimulated axons. This is particularly likely in the highly structured laminar organisation of the hippocampus, however may not be the case in more complex wiring scenarios. The fundamental difference between evoked event and mini event analysis however, is timing. With evoked events, the exact onset time is known, and detection is typically set by measuring trace deflection of a certain level at a known time point. With mini analysis, event times are random and unknown, and therefore detection must rely on fundamentally different parameters, for example template matching at any point on a recorded trace. As a result, both biological and analysis considerations are very different for minis and evoked events. We feel it is most constructive to keep the focus of this manuscript only on mini analysis, and therefore would prefer not to expand the scope to include other recording configurations.

5. The authors also do not consider the role of several recording parameters. For instance, cell capacitance, series resistance, membrane resistance as well as multi-compartment morphology of central neurons would all be factors complicating voltage/space clamp accuracy and noise.

We agree that many additional factors could influence the properties of recorded events. Series resistance is a particular concern, as it is likely to differ substantially between individual recordings. Series resistance changes would differently filter recorded events, causing differences in both amplitude and kinetics of synaptic event distributions. As demonstrated in the revised manuscript, both of these properties will affect detected event distributions. In addition to the new figure 5 (kinetic changes) which is related to this issue, we have also highlighted the impact of series resistance in the new '*Practical steps for minimising misinterpretation due to recording conditions*' section of the discussion (line 491, see lines 409-518 and 522-529). We also now report the series resistance of our recorded datasets in both the results and methods (lines 237 and 390). Regarding intrinsic cell properties, this is likely to have a lesser influence when comparing manipulations performed within a single neuronal population, however is indeed important to consider, and we have included a statement in the discussion to note the potential impact of cell properties (lines 523 and 569).

6. In addition, a fundamental reason why mini frequency may not be a good indicator of synaptic changes is that when assessed at the single synapse level, it often does not correlate with evoked release probability.

The difference in properties between spontaneous and evoked release are certainly important. We have added a short comment on the relation between minis and evoked events in the '*Biological interpretation of mPSC changes*' section of the discussion (lines 576-581).

Thanks again to the reviewer for the highly constructive comments to improve this manuscript.

Referee #2:

Overall. This article provides some important insights into the limitations of estimates of synaptic parameters from measurement of miniature PSC events. The introduction underestimates how well known many of these issues are (were) in the literature but correctly recognises the importance of raising these issues to another generation of neuroscientists; particularly those that do not fully appreciate the limitations of the single electrode whole-cell voltage clamp. The

background and appreciation of these limitations is much greater than these authors seem to acknowledge in their introduction, but perhaps this also reflects the general awareness of synaptic experimentalists of the limitations of their technique; so the focus here is timely and will help correct/improve synaptic studies for the future. It is important that the authors conduct a thorough re-write of the whole text to reduce ambiguity and make their conclusions clear to the reader (see figure legend summary sentences for instance).

Thanks to the reviewer for their positive feedback on our work, highlighting 'the importance of raising these issues', 'timely' etc. We also thank the reviewer for their extensive and constructive comments that have helped to greatly improve our manuscript. We appreciate the time and effort.

My specific points (minor and major) are listed by line number

Abstract

20. Just say extensively, delete 'one of the most', and.

We have included this suggestion.

21. Delete 'approaches'.

The abstract has been updated.

Re-write the abstract - the first half is introduction - your abstract should focus on why you have conducted the study, what you have done and provide more specific details of your results. Many of the sentences contain empty phrases.

We have now revised the abstract to better summarise the results of our analysis. We hope that this version is now more useful for potential readers.

31/32. - Sentence is confusing or poorly constructed..... Measured changes in mIPSC amplitude or frequency, falsely report changes in the synaptic parameters, due to incomplete detection of the distribution and magnitude of the synaptic events....?

The abstract has been revised to better phrase the results.

34. Delete 'not only'

This phrasing has been omitted from the revised abstract.

35. Delete 'but also' > and

This phrasing has been omitted from the revised abstract.

Introduction

The introduction needs an extensive rewrite; it currently fails to do justice to the long history of quantal analysis and does not provide a broad introduction to the subject. The last paragraph needs to make clear what data used here is in silico and which is animal experimental.

We have thoroughly revised the introduction to better present the history of quantal analysis and the path from such findings to the widespread use and interpretation of mini data (see lines 66-81). This suggestion has greatly improved the manuscript, providing better context on careful synaptic physiology for present day readers. Thanks to the reviewer for these comments. We have also now clarified the origin of our conclusions as *in silico* or experimental in the introduction (lines 129-139).

42. Delete the first sentence, and the second or find a better means of expression. We cannot know that each of trillions of synapses has unique properties!

We have rewritten the introduction and removed the sentences in question.

47. Your introduction to quantal synaptic events should be broader and can include the neuromuscular junction. Your citation in the introduction sometimes lacks depth, and includes too many reviews over original sources (e.g. Kavalali 2015; Edwards 1991).

As mentioned above, we have now extensively revised the introduction to provide far greater context to historical synaptic physiology. We hope that the revised introduction better reflects the accumulated literature, and provides a more useful reference for newcomers to the field.

64. Expand to provide information or delete this sentence. The remainder of the sentence makes statements without citation, but perhaps you will deal with these matters later in your manuscript. It would be better to cite, as your bald statements do not help the uninitiated to access your article. You do not need to cite every single thing, but there should be sufficient to satisfy the knowledgeable reader that you know the historical context whilst also providing background for those seeking an introduction to the basics.

In revising the introduction, we have expanded the discussion and citation of the points in question.

Citation:

It is surprising that you do not take a broader view of quantal synaptic physiology, nor cite or discuss the work of Redman, Edwards and Walmsley in the late 1970's and 80s. While you are working on central synapses, many of the fundamentals (and limitations) were studied at the NMJ. There are other studies of a detection limit and discussions of technical limitations. I see that you acknowledge this in the discussion, but you need not build 'straw men' for an important paper.

As mentioned above, in revising the introduction we have included greater background on quantal synaptic physiology, including a new paragraph detailing the work on quantal transmission at spinal cord neurons as suggested by the reviewer (see lines 66-79). This provides a much better introduction to how mini analysis is commonly interpreted, and the issues with this approach for understanding synaptic transmission. We also now more clearly state in the key points (line 21), abstract (line 34), and introduction (line 123) that the issues we highlight were previously acknowledged, but are now largely forgotten by those who are not so physiologically minded.

Methods:

These are normally placed after the introduction.

The manuscript format has been updated to conform to journal standards.

Why were experiments conducted at room temperature, rather than Physiological temperatures? Perhaps a note about the impact of temperature on fluctuations and noise in real recordings might be appropriate.

We have now included a sentence on the impact of temperature on recording parameters in the extended discussion of experimental parameters (line 515).

Why did you use Cs methylsulphonate as your internal ion? Are you aware that the very low internal chloride ion concentration would have generated huge junction potentials? You should state what those junction potentials were, even if you have not corrected for them. Low chloride internals can also cause problems in current passing, as silver chloride is the interface of your silver wire with the electrolyte....

This is a very important point. We have now calculated and corrected for the effect of liquid junction potentials throughout the manuscript, which was indeed a large, +10.1 mV potential. This value is particularly relevant when considering the expected magnitude of holding potential scaling, and we have included further discussion of this point in response to the reviewer's later comment (see below).

What was the mean and range of series resistances of your recording pipettes? These should not be those data measured at the start of the recording, but measured at the time of collecting a given data set (as you know series resistance often increases rapidly following the initial break-in perhaps due to plugging of the pipette lumen).

This is another important point raised. We have now reported the series resistance of both recorded datasets during data collection in the results and methods sections (lines 237 and 390). We have also added a sentence to the methods to clarify that we monitor series resistance during mPSC acquisition (line 234), and clearly highlighted the importance of series resistance comparability between datasets in the revised discussion (see line 409 and line 522).

All abbreviations need to be specified and drugs used named in full, and their source (at least in the methods).

We have crossed checked all abbreviations and included full drug names and sources in the methods section (line 223).

You need some statements about statistics.

We have included this information under the subheading '*Statistics, data analysis and visualisation*' in the methods section of the revised manuscript (line 240).

You must review and fulfil the detailed information stated in the instructions to authors about use of Animals in Research and include the requested information in your methods section. Some of the information is stated but not all.

Thank you for pointing this out, we have now included this information in the new 'Animals' section (line 202).

Figures

All figures and graph error bars need SEM to be replaced with Standard Deviation (SD).

We agree that SEM was not a clear depiction of the data structure. We have now replaced mean amplitude bar charts with box and whisker plots better representing the distribution of amplitudes. We have indicated the mean value with an overlaid symbol and reported this value in the figure legend. In addition, simulated analyses are no longer presented as multiple replicates, but a single larger dataset, as is more appropriate for modelling data. We hope that this data presentation better reflects the data structure, while maintaining clear presentation of how detection influences mean values – the readout most often used for interpreting mini data.

The first sentence of each legend should ideally give the take-home message for that figure. This nearly works for Fig 1 - but could be clearer: e.g. "The limit to detection misrepresents the event distribution by failing to include small events below the detection limit." The other figure legends need something similar.

We have reworded all figure legend introductory sentences for clearer message delivery.

Figure 1

1C Please add your calculated detection limit to the graphs, perhaps with a vertical dashed line?

We have now included the calculated limit as a vertical dashed line.

1D, 1G - Ambiguous axis titles - please change to: Mean mPSC Amp. (pA); Observed mPSC Fq. (Hz)

We have updated all amplitude charts to box and whisker format with the mean clearly depicted as an overlaid symbol, and retitled all frequency chart axes to 'observed mPSC frequency' as suggested.

Figure 2

2C, 2D and 2G. Please add your calculated detection limit to the graphs, perhaps with a vertical dashed line?

We have now included the calculated limit.

Figure 3

3D - Ambiguous axis titles - please change to: Mean mPSC Amp. (pA); Observed mPSC Fq. (Hz)

We have updated these charts as noted above.

Figure 4.

4C - Ambiguous axis titles - please change to: Mean mPSC Amp. (pA); Observed mPSC Fq. (Hz)

We have updated these charts as noted above.

Legend. You might measure or determine the detection limit.... You do not detect the detection limit.

Correct - we have updated all relevant phrasing to 'estimation', which is more appropriate for this method (lines 27, 45, 54, 137, 194, 394, 413, 417, 473 etc.).

Figure 5 Diagram.

This contains some useful information in the table but there are issues such as vesicle recycling and the interaction of voltage-clamp with synaptic location which are omitted.

We have now included both of these factors in the table and schematic depiction.

Results

92. This phrase reads well but is not very scientific 'establishing a pipeline for simulating mPSC recordings', please state that simulated mPSCs were generated as detailed in the methods.

We have reworded this sentence as suggested by the reviewer.

121. 'by at least a factor of two'.

This error has been corrected.

186. to assess the unreliability > to assess the reliability. Seems like a less biased statement?

We agree with the reviewer and have made the suggested change.

195. Using driving force to change synaptic amplitude has at least two important pitfalls under voltage-clamp conditions. The fact that these issues may have caused an underestimation of the holding potential is implied by the relatively small measured changes in the mPSC amplitude (Fig 4B; looking only at those events larger than the detection limit of 5.9pA). As you state in your text, I would have expected an increase of between 25-33% on changing the driving force from -60 to -80 mV (assuming a reversal potential of 0mV for an excitatory glutamatergic synapse). Your measured mPSCs increase by much less than this (more like 10-15%) in 4B. While you take this to support your hypothesis, it is as likely that you have not achieved your command voltage change (from -60 to -80 mV) at the active synapses due to (1) voltage drop across the patch pipette series resistance (2) decay of your command voltage due to cable theory and the neuron length constant (i.e. that distance between your pipette at the soma and the synapse on the dendrites). While your use of internal Cs will have raised your length constant - to your advantage; your use of low chloride has introduced a large uncompensated junction potential; so you never were at -60mV holding potential to start with.....This could further undermine the accuracy of your voltage command. So you really need to know the value of your Junction potential. It would be particularly interesting if you were to apply your methods to data collected from preparations in which synapses are made only within a highly confined/proximal location to the recording site - for instance in an 'old fashioned' dissociated hippocampal preparation (where synaptic boutons remain attached to the soma and spontaneously release minis) or in a preparation such as the calyx of Held where most excitatory synapses form on the soma.

Thanks for this serious and thoughtful reflection on our work – these are very important considerations. The reviewer is correct to highlight the effect of the junction potential on our analysis. We have now calculated the liquid junction potential between our solutions, and as expected this has a large contribution, +10.1 mV. Accounting for this value, our corrected recording potentials are -70 and -90 mV. Using these values, we would predict a 1.29 rather than 1.33 times increase in event amplitudes caused by the increased driving force. We have updated both manuscript and figure (Figure 6 of the revised manuscript) to use junction potential corrected values, and an expected 1.29x scaling factor. This results in a minor change in the estimated detection limit, from 5.9 to 6.2 pA.

Regarding the size of the expected change, we reanalysed our recorded dataset, comparing the event distributions recorded at -80 mV command potential, to the -60 mV command potential dataset scaled either 1.33 times (uncorrected scaling), 1.29 times (junction corrected), or 1.1 times (10% increase suggested by the reviewer) (see response to reviewers Figure 1, below). For large events, away from the influence of the detection limit, event histograms with ~1.3x scaling were overlapping with our -80 mV dataset, while 1.1 times scaling was not. Therefore our data are consistent with having effectively induced 1.3 times, not 1.1 times amplitude scaling on the recorded synaptic population. While we would of course not expect perfect voltage clamp control of synaptic currents, and we now explicitly state the issue of space-clamp in the revised manuscript (lines 462 & 475), given the congruence between expected and recorded amplitude scaling, recorded events appear to be sufficiently well manipulated for use of this approach to estimate the detection limit. We agree with the reviewer that it would be theoretically of interest to study well controlled soma proximal synaptic datasets, however the vast majority of mini data analysis is performed on complex neurons. Given that the focus of this manuscript is to aid robust mini analysis under such conditions, model system experiments would not add so much to the utility of this manuscript.

Thanks again to the reviewer for raising these issues. Revising the manuscript to include consideration of liquid junction potentials has substantially increased our accuracy and validity.

Response to reviewers Figure 1. Scaling our experimentally recorded dataset (-60 mV command potential) with either a 10, 29 or 33% increase in event amplitude shows alignment with the -80 mV dataset for ~30% but not 10% amplitude increase. Therefore holding potential induced scaling is more consistent with a 30 than 10% experimental effect.

211. Fallibility. I would suggest that at least part of the overall problem is the quality of voltage-clamp and of course this is as big a problem for you as it is for other investigators. It need not be a terminal problem, as you could incorporate this aspect of the solution into your manuscript with a re-write. Indeed, I would also suggest that the value of your manuscript would be increased if you were to present methods by which you could estimate, compensate or place limits on the quality of experimental data.

We agree with the reviewer that recording quality is of critical importance, in particular, having equivalent recording quality between experimentally compared conditions. We have now emphasised this in the revised manuscript by reporting the series resistance of our recorded data in the results and methods (lines 237 and 390), explicitly stating the problem of space-clamp (line 475), and by expanding the discussion to include a new '*Practical steps for minimising misinterpretation due to recording conditions*' section to the discussion (line 491). We have also now included a step-by-step plan for considering the limitations highlighted in our study, including the quality of experimental data (lines 519-555). This is not meant as a comprehensive protocol, but we hope increases the value of the manuscript as a resource for better mini analysis.

427. In the past the Journal of Physiology requested confirmation that those acknowledged as having provided feedback on a manuscript had given permission for their names to be used.

All those acknowledged for their contribution to this manuscript have now confirmed their permission to be named.

We would like to thank both reviewers again for their comments, which have certainly made this manuscript a more comprehensive resource for consideration of mini datasets.

Dear Dr Watson,

Re: JP-RP-2025-288183R1 "'Mini analysis' misrepresents changes in synaptic properties due to incomplete event detection" by Ingo Harald Greger and Jake F Watson

Thank you for submitting your manuscript to The Journal of Physiology. It has been assessed by a Reviewing Editor and by 2 expert referees and we are pleased to tell you that it is acceptable for publication following satisfactory revision.

REVISION CHECKLIST:

We look forward to receiving your revised submission.

Yours sincerely,

Katalin Toth
Senior Editor
The Journal of Physiology

REQUIRED ITEMS

- You must start the Methods section with a paragraph headed Ethical approval (https://jp.msubmit.net/cgi-bin/main.plex?form_type=display_requirements#methods).

Research must comply with The Journal's policies regarding animal experiments (<https://physoc.onlinelibrary.wiley.com/hub/animal-experiments>) and adherence to these policies must be stated in the manuscript.

Authors should confirm in their Methods section that their experiments were carried out according to the guidelines laid down by their institution's animal welfare committee, including an ethics approval reference number. The Methods section must contain a statement about access to food, water and housing, details of the anaesthetic regime: anaesthetic used, dose and route of administration, and method of killing the experimental animals.

EDITOR COMMENTS

Reviewing Editor:

Thank you for your effort to address all the reviewers concerns. Both reviewers commend your revisions. Before final acceptance, please ensure to include the paragraph on the detection of evoked events into the main manuscript as reviewer 1 suggests. There are also a couple other small improvements suggested by reviewer 2 for you to incorporate into the final manuscript. Please also declare whether the decapitation was conducted with or without anesthesia.

REFEREE COMMENTS

Referee #1:

The authors have diligently and candidly addressed my earlier comments. I do not see major issues that preclude publication. That said, I strongly recommend adding the section on "implications for detection of evoked events" (in response to my question #4) to the main body of the paper.

Referee #2:

The authors have made substantial improvements to the manuscript and largely addressed the issues raised in my previous

report. The figures are clear and the textual changes help the reader understand the complexities involved in this kind of analysis.

On re-reading I found a few minor ambiguities - especially in the newly added sections; so the authors need to give the MS a thorough proof read to make sure the meaning of their text is clear.... You know what you mean, but your reader might not.....

Some examples:

Ln 535 - Insert mPSC events.

Ln 538 - Delete > "while uncomfortable". I assume you mean that the recommended process implies some data selection, but so long as this aspect of the method is explicitly declared, then this is acceptable. You could expand the explanation, it's up to you.

Ln 541 - > recording duration (not 'length' - I thought that you were discussing the length constant at first).

Ln 544 > mPSC frequency

There are probably other similar minor issues, elsewhere in the MS, that I have missed.

END OF COMMENTS

JP-RP-2024-288183R1 Point-by-point response to reviewers

Reviewing Editor:

Thank you for your effort to address all the reviewers concerns. Both reviewers commend your revisions. Before final acceptance, please ensure to include the paragraph on the detection of evoked events into the main manuscript as reviewer 1 suggests. There are also a couple other small improvements suggested by reviewer 2 for you to incorporate into the final manuscript. Please also declare whether the decapitation was conducted with or without anesthesia.

We are grateful to the editor and referees for their feedback on our manuscript. We now provide a revised manuscript file including the requested changes for both reviewer 1 and 2, and including the required information on animal termination.

Referee #1:

The authors have diligently and candidly addressed my earlier comments. I do not see major issues that preclude publication. That said, I strongly recommend adding the section on "implications for detection of evoked events" (in response to my question #4) to the main body of the paper.

Thanks to the reviewer for their positive comments on our revision. We have now included the requested information on evoked events in the revised manuscript (lines 468-471 of the merged PDF).

Referee #2:

The authors have made substantial improvements to the manuscript and largely addressed the issues raised in my previous report. The figures are clear and the textual changes help the reader understand the complexities involved in this kind of analysis.

Thanks to the reviewer for their positive feedback.

On re-reading I found a few minor ambiguities - especially in the newly added sections; so the authors need to give the MS a thorough proof read to make sure the meaning of their text is clear.... You know what you mean, but your reader might not.....

We have now rechecked and corrected minor errors in the revised manuscript.

Some examples:

Ln 535 - Insert mPSC events.

We have made the suggested change.

Ln 538 - Delete > "while uncomfortable". I assume you mean that the recommended process implies some data selection, but so long as this aspect of the method is explicitly declared, then this is acceptable. You could expand the explanation, it's up to you.

We have made the suggested change.

Ln 541 - > recording duration (not 'length' - I thought that you were discussing the length constant at first).

We have made the suggested change.

Ln 544 > mPSC frequency

There are probably other similar minor issues, elsewhere in the MS, that I have missed.

We have made the suggested change.

Dear Dr Watson,

Re: JP-RP-2025-288183R2 "'Mini analysis' misrepresents changes in synaptic properties due to incomplete event detection" by Ingo Harald Greger and Jake F Watson

We are pleased to tell you that your paper has been accepted for publication in The Journal of Physiology.

Yours sincerely,

Katalin Toth
Senior Editor
The Journal of Physiology

If you would like to receive our 'Research Roundup', a monthly newsletter highlighting the cutting-edge research published in The Physiological Society's family of journals (The Journal of Physiology, Experimental Physiology, Physiological Reports, The Journal of Nutritional Physiology and The Journal of Precision Medicine: Health and Disease), please click this link, fill in your name and email address and select 'Research Roundup':

<https://www.physoc.org/journals-and-media/membernews>

- **TRANSPARENT PEER REVIEW POLICY:** To improve the transparency of its peer review process, The Journal of Physiology publishes online as supporting information the peer review history of all articles accepted for publication. Readers will have access to decision letters, including Editors' comments and referee reports, for each version of the manuscript as well as any author responses to peer review comments. Referees can decide whether or not they wish to be named on the peer review history document.
- You can help your research get the attention it deserves! Check out Wiley's free Promotion Guide for best-practice recommendations for promoting your work at: www.wileyauthors.com/eeo/guide. You can learn more about Wiley Editing Services which offers professional video, design, and writing services to create shareable video abstracts, infographics, conference posters, lay summaries, and research news stories for your research at: www.wileyauthors.com/eeo/promotion.
- **IMPORTANT NOTICE ABOUT OPEN ACCESS:** To assist authors whose funding agencies mandate public access to published research findings sooner than 12 months after publication, The Journal of Physiology allows authors to pay an Open Access (OA) fee to have their papers made freely available immediately on publication.

EDITOR COMMENTS

Reviewing Editor:

Thank you for taking care of the last revisions, and congratulations on an interesting paper.